# Ignoring carbon emissions from thermokarst ponds results in overestimation of tundra net carbon uptake

Lutz Beckebanze[1,2,*], Zoé Rehder[3,4,*], David Holl[1,2], Christian Wille[5], Charlotta Mirbach[1,2], and Lars Kutzbach[1,2]

[1]Institute of Soil Science, Universität Hamburg, Germany
[2]Center for Earth System Research and Sustainability (CEN), Universität Hamburg, Germany
[3]Department of the Land in the Earth System, Max Planck Institute for Meteorology, Hamburg, Germany
[4]International Max Planck Research School on Earth System Modeling, Hamburg, Germany
[5]Helmholtz-Zentrum Potsdam – Deutsches Geo Forschungs Zentrum (GFZ), Potsdam, Germany
[*]These authors contributed equally to this work.

**Correspondence:** Lutz Beckebanze (lutz.beckebanze@uni-hamburg.de), Zoé Rehder (zoe.rehder@mpimet.mpg.de)

**Abstract.** Arctic permafrost landscapes have functioned as a global carbon sink for millennia. These landscapes are very heterogeneous, and the omnipresent water bodies within them act as a carbon source. Yet, few studies have focused on the impact of these water bodies on the landscape carbon budget. We deepen our understanding of carbon emissions from thermokarst ponds and constrain their impact by comparing carbon dioxide and methane fluxes from these ponds to fluxes from the surrounding tundra. We use eddy covariance measurements from a tower located at the border between a large pond and semi-terrestrial tundra.

When we take the open-water areas of thermokarst ponds into account, our results show that the estimated summer carbon uptake of the polygonal tundra is 11% lower. Further, the data show that open-water methane emissions are of similar magnitude as polygonal tundra emissions. However, some parts of the pond's shoreline exhibit much higher emissions. This finding underlines the high spatial variability of methane emissions. We conclude that gas fluxes from thermokarst ponds can contribute significantly to the carbon budget of arctic tundra landscapes. Consequently, changes in the water body distribution of tundra landscapes due to permafrost degradation may substantially impact the overall carbon budget of the Arctic.

## 1 Introduction

Water bodies make up a significant part of the arctic lowlands with an areal coverage of about 17 % (Muster et al., 2017), and act as an important carbon source in a landscape that is an overall carbon sink (Kuhn et al., 2018). Intensified permafrost thaw in the warming Arctic will change the distribution of water bodies, and thereby change their contribution (Andresen and Lougheed, 2015; Bring et al., 2016) to the landscape carbon budget (Kuhn et al., 2018) of tundra landscapes. However, data on greenhouse gas emissions from arctic water bodies are still sparse, especially data with high temporal resolution and from non-Yedoma regions (Vonk et al., 2015).

Our study site in the Lena River Delta, Siberia, is located on an island mostly characterized by non-Yedoma polygonal tundra (Fig. 1). This landscape features many ponds; we define ponds as water bodies with an area of less than $8 \cdot 10^4$ m², following

Ramsar Convention Secretariat (2016); Rehder et al. (2021). Within our area of interest, ponds cover about the same area as lakes (Abnizova et al., 2012; Muster et al., 2012). The ponds on Samoylov Island have formed almost exclusively through thermokarst processes: The soil has a high ice content, so when the ice melts, the ground subsides, and thermokarst ponds form

(Ellis et al., 2008). These thermokarst ponds are often only as large as one polygon (polygonal ponds). When several polygons are inundated, this can cause larger shallow thermokarst ponds to form, which we term merged polygonal ponds (Rehder et al., 2021). Holgerson and Raymond (2016) as well as Wik et al. (2016) report that ponds emit more greenhouse gases per unit area than lakes, defined here as water bodies with an area larger than $8 \cdot 10^4$ m$^2$. Thus, in our study area, they have a greater potential than lakes to counterbalance the carbon uptake of the surrounding tundra (McGuire et al., 2012; Jammet et al., 2017; Kuhn

et al., 2018). To better understand the impact of thermokarst ponds on the landscape carbon flux, we compare carbon dioxide ($CO_2$) and methane ($CH_4$) fluxes from thermokarst ponds to fluxes from the semi-terrestrial tundra. The semi-terrestrial tundra consists of wet and dry tundra, and overgrown shallow water, which are the terrestrial land-surface types used by Muster et al. (2012) to classify Samoylov Island.

The main geophysical and biochemical processes that drive $CH_4$ fluxes are different to the ones that drive $CO_2$ fluxes.

The microbial decomposition of dissolved organic carbon, which is introduced laterally into the aquatic system through rain and meltwater (Neff and Asner, 2001), dominates aquatic $CO_2$ production. When supersaturated with dissolved $CO_2$, ponds emit $CO_2$ into the atmosphere through diffusion. While photosynthetic $CO_2$ uptake has been observed in some clear arctic water bodies (Squires and Lesack, 2003), most arctic water bodies are net $CO_2$ sources (Kuhn et al., 2018). Estimates of $CO_2$ emissions range from close to zero (0.028 g m$^2$ d$^{-1}$ by Treat et al. (2018), and 0.059 g m$^2$ d$^{-1}$ by Jammet et al. (2017)) to

substantial (1.4–2.2 g m$^2$ d$^{-1}$ by Abnizova et al. (2012)).

Within just one site, $CH_4$ emissions from a water body can vary by up to five orders of magnitude: 0.5–6432 mg m$^2$ d$^{-1}$ (Bouchard et al., 2015). The $CH_4$ that ponds emit is mostly produced in sub-aquatic soils and anoxic bottom waters (Conrad, 1999; Hedderich and Whitman, 2006; Borrel et al., 2011). Additionally, $CH_4$ might also be produced in the oxic water column (Bogard et al., 2014; Donis et al., 2017), though this location of methanogenesis is only significant in large water

bodies (Günthel et al., 2020). Moreover, there is still ongoing debate as to whether methanogensis occurs in oxic waters at all (Encinas Fernández et al., 2016; Peeters et al., 2019). $CO_2$ is also formed as a byproduct of the methanogensis process (Hedderich and Whitman, 2006). Water bodies emit $CH_4$ produced in their benthic zone through diffusion, ebullition (sudden release of bubbles), or plant-mediated transport. The varying contributions of these three local methane emissions pathways lead to high spatial variability between water bodies and within a single water body (Sepulveda-Jauregui et al., 2015; Jansen

et al., 2019). In particular, local seep ebullition causes high spatial variance of $CH_4$ emissions within one water body (Walter et al., 2006). Variability in the coverage and composition of vascular plant communities in a water body can also increase $CH_4$ variability because $CH_4$ transport efficiency can be species-specific (Knoblauch et al., 2015; Andresen et al., 2017).

To study spatial and temporal patterns of carbon emissions from thermokarst ponds, we analyzed land-atmosphere $CO_2$ and $CH_4$ flux observations from an eddy covariance (EC) tower on Samoylov Island, Lena River Delta, Russia. We set up the

EC tower within the polygonal tundra landscape at the border between a large merged polygonal pond and the surrounding semi-terrestrial tundra for two months in summer 2019. The polygonal structures were still clearly visible along the shore

and underwater, and most of the pond was shallow (Rehder et al., 2021). Due to the tower's position, fluxes from the merged polygonal pond were the dominant source of the observed EC fluxes under easterly winds. From other wind directions, the observed EC fluxes were dominated by semi-terrestrial polygonal tundra with only a low influence from small polygonal ponds.

This paper aims to deepen the understanding of carbon emissions from thermokarst ponds and constrain their impact on the landscape carbon balance. We (1) examine the temporal and spatial patterns of NEE and the spatial pattern of $CH_4$ flux from semi-terrestrial tundra and thermokarst ponds, and (2) investigate the influence of the thermokarst ponds on the landscape NEE of $CO_2$ during the months June to September 2019. To this end, we use a footprint model and model net ecosystem exchange (NEE) of $CO_2$ using the footprint weights of semi-terrestrial tundra and thermokarst ponds.

## 2 Methods

### 2.1 Study site

Samoylov Island (72°22'N, 126°28'E) is located in the southern part of the Lena River Delta (Fig. 1, b). It is approximately five $km^2$ large and consists of two geomorphologically different components. The western part of the island ($\sim$2 $km^2$) is a floodplain, which is flooded annually during the spring. The eastern part of the island ($\sim$3 $km^2$), a late-Holocene river

terrace, is characterized by polygonal tundra. The partially degraded polygonal tundra at this study site features high spatial heterogeneity on a scale of a few meters in several aspects, including vegetation, water table height, and soil properties. Dry and wet vegetated parts of the semi-terrestrial tundra are interspersed with small and large thermokarst ponds (<1–>10,000 $m^2$) and with larger lakes (up to 0.05 $km^2$, Boike et al. (2015a); Kartoziia (2019)). The island is surrounded by the Lena River and sandy floodplains, creating additional spatial heterogeneity on a larger scale.

This study focuses on a merged polygonal pond (Fig. 1, d, and A1) on the eastern part of the island. This merged polygonal pond has a size of 0.024 $km^2$ with a maximum depth of 3.4 meters and a mean depth of 1.2 meters (Rehder et al., 2021; Boike et al., 2015a). In an aerial image of the pond, the polygonal structures are clearly visible under the water's surface (Boike et al., 2015c). The vegetated shoreline of this merged polygonal pond is dominated by *Carex aquatilis*, but it also features *Carex chordorrhiza*, *Potentilla palustris*, and *Aulacomnium spp.*. These plants grow in the water near the shore while the deeper parts

of the merged polygonal pond are vegetation-free.

### 2.2 Instruments

We measured gas fluxes using an eddy covariance (EC) tower between July 11 and September 10, 2019. The EC tower was located on the eastern part of Samoylov Island, directly at the western shore of the merged polygonal pond (Fig. 1, d). The EC instruments were mounted on a tripod at a height of 2.25 meters (Fig. A1). The tower was equipped with an enclosed-path

$CO_2$/$H_2O$ sensor (LI-7200, LI-COR Biosciences, USA), an open-path $CH_4$ sensor (LI-7700, LI-COR Biosciences, USA), and a 3D-ultrasonic anemometer (R3-50, Gill Instruments Limited, UK). All instruments had a sampling rate of 20 Hz. We also installed radiation-shielded temperature and humidity sensors at the EC tower (HMP 155, Vaisala, Finland) and used data from

a photosynthetically active radiation (PAR) sensor mounted on a tower approximately 500 meters to the west of the EC tower (SKP 215, Skye Instruments, UK). Additional meteorological data for Samoylov Island was provided by Boike et al. (2019).

## 2.3 Data processing

We performed the raw data processing and computation of half-hourly fluxes for open-path and enclosed-path fluxes ($CO_2$, $CH_4$ and $H_2O$) using *EddyPro 7.0.6* (LI-COR, 2019). The convention of this software is that positive fluxes are fluxes from the surface to the atmosphere, while negative fluxes indicate a flux from the atmosphere downwards. Raw data screening included spike detection and removal according to Vickers and Mahrt (1997) (1 % maximum accepted spikes and a maximum of three consecutive outliners). Additionally, we applied statistical tests for raw data screening, including tests for amplitude resolution, skewness and kurtosis, discontinuities, angle of attack, and horizontal winds steadiness. All of these tests' parameters were set to *EddyPro* default values. We rotated the wind-speed axis to a zero-mean vertical wind speed using Kaimal and Finnigan's (1994) "double rotation" method. Further, we applied linear de-trending to the raw data following Gash and Culf (1996) before performing flux calculations. We compensated time lags via automatic time lag optimization using a time lag assessment file from a previous EddyPro run. In this previous time lag assessment, the time lags for all gases were detected using covariance maximization (Fan et al., 1990), resulting in time lags between 0–0.4 s for $CO_2$ and -0.5– +0.5 s for $CH_4$. For $H_2O$, the time lag was humidity-dependent and was calculated for 10 humidity classes. We compensated for air-density fluctuations due to thermal expansion and contraction and varying water-vapor concentrations, following Webb et al. (1980). This correction depends on accurate measurements of the latent and sensible heat flux and was applied to the open-path data of the LI-7700. For the LI-7700 in particular, the correction term can be larger than the flux itself, but the correction was derived from the underlying physical equations. Because we used well-calibrated instruments as well as *EddyPro*, which uses an up-to-date implementation of the correction, we were confident that the LI-7700 would provide accurate $CH_4$ flux estimates. For enclosed-path data, we performed a sample-by-sample conversion into mixing ratios to account for air density fluctuations (Ibrom et al., 2007b; Burba et al., 2012). Flux losses occurred in the low- and high frequency spectral range due to different filtering effects. We compensated flux losses in the low-frequeny range in accordance with Moncrieff et al. (2004) and in the high-frequency range in accordance with Fratini et al. (2012). For the high-frequency range compensation method, a spectral assessment file was created using Ibrom et al.'s (2007a) method. The spectral assessment resulted in cut-off frequencies of 3.05 Hz and 1.67 Hz for $CO_2$ and $CH_4$, respectively. For $H_2O$, we found a humidity-dependent cut-off frequency between 1.25 Hz (RH 5–45 %) and 0.21 Hz (RH 75–95%). We performed a quality check on each half-hourly flux following the 0-1-2 system proposed by Mauder and Foken (2004). In this quality check, flux intervals with the lowest quality received the flag "2" and were excluded from further analysis.

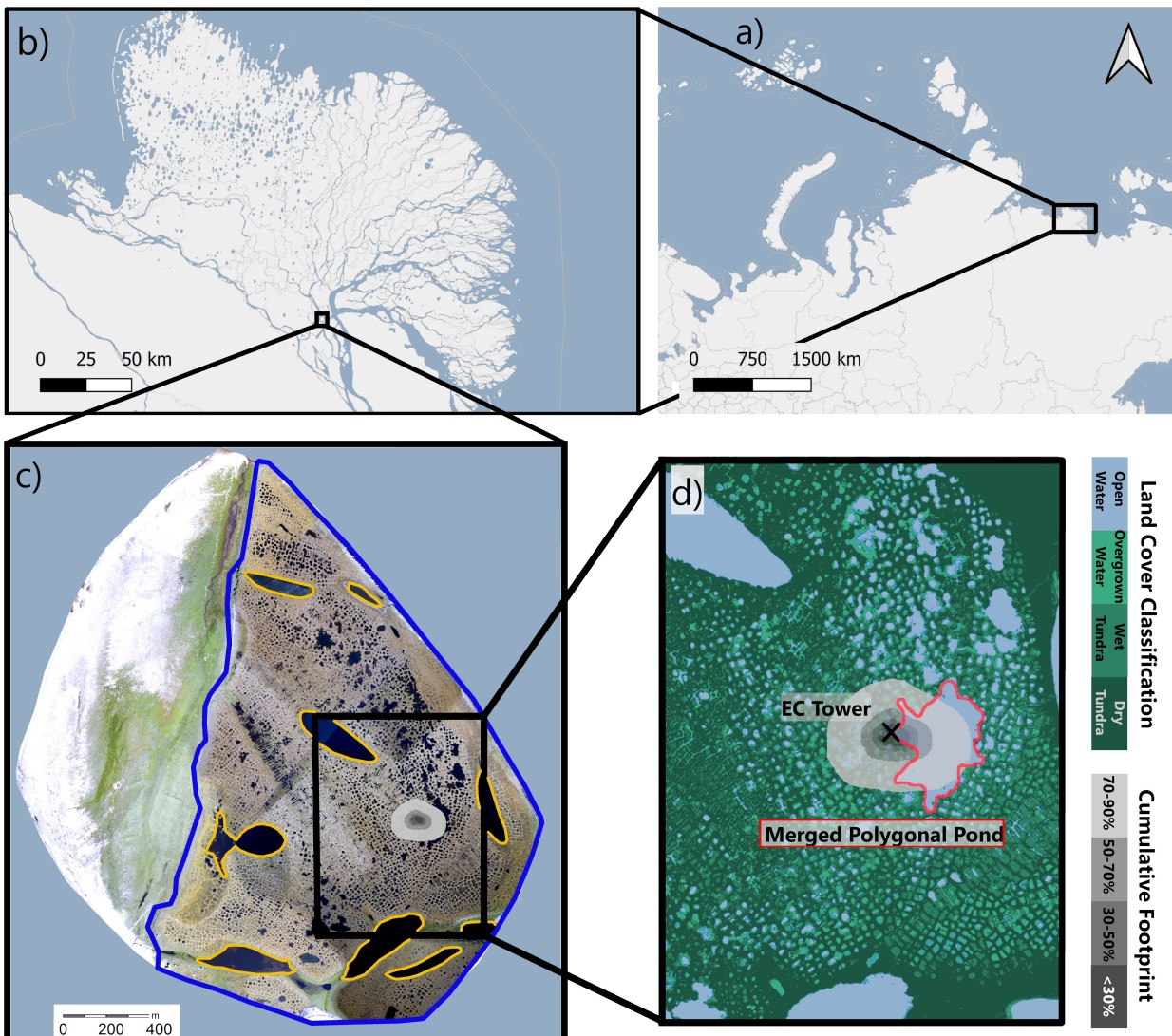

**Figure 1.** The location of the study site in Russia is shown in (a) and the location of Samoylov Island within the Lena River Delta is shown in (b). Samoylov Island is shown in (c); the surrounding Lena River appears in light blue. The outline of the river-terrace land-cover classification (Sect. 2.4.1) is indicated by the blue line. We focus on the polygonal tundra, however, large lakes are excluded (circled in yellow). In (d), the land-cover classification is drawn in blue (open water) and green shades (dark green: dry tundra; medium green: wet tundra; and light green: overgrown water). The merged polygonal pond studied here is outlined in red. The location of the EC tower is marked by a black cross. The cumulative footprint (see Sect. 2.4.2) is shown in gray shades. 30 % of the flux likely originated from within the dark gray area, 50 % from within the medium-dark gray area, 70 % from within the medium-light gray area and 90 % from within the light gray area. Map data from © OpenStreetMap contributors 2020, distributed under the Open Data Commons Open Database License (ODbL) v1.0 (a & b) and modified based on Boike et al. (2012) (c & d).

## 2.4 Data analysis

### 2.4.1 Land-cover classification

The land-cover classification covers the late-Holocene river terrace of Samoylov Island (3.0 km$^2$, area within the blue line
in Fig. 1, c). It is based on high-resolution near-infrared (NIR) orthomosaic aerial imagery obtained in the summer of 2008
(Boike et al., 2015b). We used a subset of Muster et al.'s (2012) existing classification as a training dataset to perform a semi-
supervised land-cover classification using the *maximum likelihood algorithm* in ArcMap Version 10.8 (ESRI Inc, USA). We
then applied the ArcMap *majority filter* tool to the new classification. The land-cover classification has a resolution of 0.17 m
x 0.17 m,. It is projected onto WGS 1984 UTM Zone 52N and the land-cover classes include *open water* (15.7 %), *overgrown
water* (7.0 %), *dry tundra* (65.1 %), and *wet tundra* (12.1 %), as defined by Muster et al. (2012). We summarize the classes
*overgrown water*, *dry tundra*, and *wet tundra* in the land-cover type, semi-terrestrial tundra. The river terrace consists of this
semi-terrestrial tundra, large lakes, and thermokarst ponds. Since small ponds are an integral part of the polygonal tundra, we
use the term "polygonal tundra" to refer to the area of the river terrace covered by semi-terrestrial tundra and by thermokarst
ponds.

### 2.4.2 Footprint model

In deploying an EC measurement tower, the tower's location and sensor height are crucial parameters. A lower measurement
height results in a smaller footprint. The tower's footprint describes the source area of the flux within the surrounding landscape.
As we installed sensors at a height of 2.25 m next to the merged polygonal pond, we expected to observe substantial flux signals
from the adjacent water body as well as from the surrounding semi-terrestrial tundra. Each land-cover type's contribution to
the flux signal depended on the wind direction and turbulence characteristics. We implemented the analytical footprint model
proposed by Kormann and Meixner (2001) in Matlab 2019b (MATLAB, 2019). We combined the footprint model with land-
cover classification data described in Sect. 2.4.1 to estimate the contribution of each land-cover type to each half-hourly flux
(from now on referred to as the weighted footprint fraction). The model accounted for the stratification of the atmospheric
boundary layer and required a height-independent crosswind distribution and horizontal homogeneity of the surface. The input
data required stationarity of atmospheric conditions during the flux intervals of 30 minutes.

We derived the vertical power-law profiles for the eddy diffusivity and the wind speed for each 30-minute flux depending
on the atmospheric stratification (see Eq. 6 in Kormann and Meixner (2001)). We used an analytical approach to find the
closest Monin-Obukhov (M-O) similarity profile (see Eq. 36 in Kormann and Meixner (2001)). Next, we calculated a two-
dimensional probability density function of the source area for each flux (from Eq. 9 and 21 in Kormann and Meixner (2001)).
We combined each probability density function with the land-cover classification of Samoylov Island's river terrace with its
four land-cover types (see Sect. 2.4.1). The resolution of the footprint model was set to the land-cover classification resolution
of 0.17 m x 0.17 m. Hence, we were able to estimate how much a given grid cell contributed to each 30-min flux. We also knew
each grid cell's dominant land-cover type from the land-cover classification. We combined both pieces of information for each
grid cell and calculated the sum of the fraction fluxes within the source area for each of the four land-cover types (*dry tundra*,

*wet tundra*, *overgrown water*, and *open water*) and determined the contribution of each land-cover type in respect of each 30-minute flux ($a_{dry\ tundra}$, $a_{wet\ tundra}$, $a_{overgrown\ water}$, and $a_{open\ water}$). We refer to this contribution of each land-cover type as the *weighted footprint fraction*.

We also summed all 30-min two-dimensional probability density functions over the entire deployment time. This sum is referred to as the cumulative footprint (gray shaded area in Fig. 1, c–d).

### 2.4.3  Gap-filling the $CO_2$ flux

To gap-fill the net-ecosystem exchange (NEE) fluxes of $CO_2$, we used the *bulk-NEE model* proposed by Runkle et al. (2013). The model is specifically designed to model NEE in arctic regions: It takes impacts of the polar day into account by allowing both respiration and photosynthesis to occur simultaneously throughout the day. The *bulk-NEE model* uses the sum of total ecosystem respiration (TER) and gross primary production (GPP) to describe NEE, our target variable:

$$NEE = \text{TER} + \text{GPP} \tag{1}$$

where TER and GPP have the unit µmol m$^{-2}$ s$^{-1}$. TER is approximated as an exponential function of air temperature $T_{air}$:

$$\text{TER} = R_{base} \cdot Q_{10}^{\frac{T_{air} - T_{ref}}{\gamma}} \tag{2}$$

where $T_{ref} = 15\ °C$ and $\gamma = 10\ °C$ are constant, independent parameters. $R_{base}$ (µmol m$^{-2}$ s$^{-1}$) describes the basal respiration at the reference temperature $T_{ref}$ and $Q_{10}$ (dimensionless) describes the sensitivity of ecosystem respiration to air
temperature changes.

GPP is described as a rectangular hyperbolic function of PAR (µmol m$^{-2}$ s$^{-1}$):

$$\text{GPP} = -\frac{P_{max} \cdot \alpha \cdot \text{PAR}}{P_{max} + \alpha \cdot \text{PAR}} \tag{3}$$

where $\alpha$ (µmol µmol$^{-1}$) is the initial canopy quantum use efficiency (slope of the fitted curve at PAR$= 0$) and $P_{max}$ (µmol m$^{-2}$ s$^{-1}$) is the maximum canopy photosynthetic potential for PAR $\rightarrow \infty$.

The parameters $R_{base}$, $Q_{10}$, $P_{max}$, and $\alpha$ were fitted simultaneously. To account for seasonal changes in plant physiology, we fitted the parameters for running five-day windows as proposed in Holl et al. (2019).

We split the datasets into training (70 %) and validation (30 %) data sets to test model performance. We implemented the *bulk-NEE model* in Matlab 2019b (MATLAB, 2019) using the *fit* function with the *NonLinearLeastSquares* fitting method. We used the *coeffvalues*-function to estimate the four parameters ($R_{base}$, $Q_{10}$, $P_{max}$, and $\alpha$) and the *confint*-function to estimate their
95 % confidence bounds. All partitioned fluxes were converted into $CO_2$-C fluxes in the unit g m$^{-2}$ d$^{-1}$ before data analysis.

### 2.4.4  Separating $CO_2$ fluxes from semi-terrestrial tundra and water bodies

We wanted to extract fluxes from thermokarst ponds and semi-terrestrial tundra to analyze the influence of thermokarst ponds on the carbon balance of a polygonal tundra landscape. However, due to the strong heterogeneity of the landscape and the

relatively small size of the merged polygonal pond compared to the EC footprint, we measured a mixed signal from all wind
directions. In other words, each flux that was measured with the EC method contained information from different land-cover
types. We divided the footprint into two classes – semi-terrestrial tundra and thermokarst ponds – to assess the impact of
thermokarst ponds on the carbon balance.

Similar approaches of analyzing heterogeneous eddy covariance fluxes in arctic environments have been conducted for $CO_2$
and $CH_4$ (e.g. Rößger et al., 2019a,b; Tuovinen et al., 2019). Rößger et al. (2019a,b) extracted $CO_2$ and $CH_4$ fluxes from
two different land-cover classes on a floodplain, while Tuovinen et al. (2019) separated $CH_4$ fluxes from nine individual land-
cover classes, including water, and combined them into four source classes (with no separate class for water). All three studies
differentiate between fluxes from different vegetation types. Our method is dedicated to distinguishing between fluxes from
semi-terrestrial tundra and water bodies.

To estimate $CO_2$ fluxes from the merged polygonal pond ($F_{pond}$), we first fitted the *bulk-NEE model* to training data, exclud-
ing fluxes from the direction of the merged polygonal pond ($30° < WD < 150°$). We obtained a dataset consisting of information
about as much semi-terrestrial tundra as possible. We performed this step since we expected little to no photosynthetic activity
in the open-water part of the merged polygonal pond. This gap-filled $CO_2$ flux (hereinafter $F_{modeled,mix}$) represents the polyg-
onal tundra surrounding the EC tower, meaning the flux is dominated by semi-terrestrial tundra, but also includes polygonal
ponds from wind directions of north, west, and south. In the model input, we excluded 30-minute $CO_2$ fluxes with an absolute
value of more than $4$ g m$^{-2}$ d$^{-1}$. In 38 five-day windows, we found an $R^2$ above 0.9 between the model output and the vali-
dation set. In 18 cases, we obtained an $R^2$ between 0.8–0.9; in six instances, we obtained an $R^2$ below 0.7. The final RMSE
between the model input and the gap-filled NEE had a value of $0.29$ g m$^{-2}$ d$^{-1}$.

We assumed that the total observed flux was a linear combination of the fluxes from the land-cover types weighted by their
respective contribution to the footprint. Thus, we postulated that the observed $CO_2$ flux ($F_{obs,mix}$, not gap-filled) was the
sum of the individual land-cover type fluxes ($F_{modeled,mix}$ and the merged polygonal pond $F_{pond}$) each multiplied with their
weighted footprint fraction ($a_{mix}$ and $a_{pond}$), with $a_{open\ water} = a_{pond}$, $a_{mix} = a_{sum} - a_{pond}$, and $a_{sum}$ being the sum over
all land-cover classes:

$$F_{obs,mix} = a_{pond} \cdot F_{pond} + a_{mix} \cdot F_{modeled,mix}$$
$$\Leftrightarrow F_{pond} = \frac{F_{obs,mix} - a_{mix} \cdot F_{modeled,mix}}{a_{pond}} \tag{4}$$

To improve data quality, we excluded 30-min fluxes of $F_{pond}$ when $a_{pond} < 50$ %. Then, we used the median of $F_{pond}$ for
further calculations, and we assumed that all thermokarst ponds in the EC footprint emitted the same amount of $CO_2$.

As mentioned above, the observed $CO_2$ flux from the wind direction of north, west, and south ($F_{obs,mix}$) was influenced by
polygonal ponds to a small degree. Since our aim was to assess the impact of thermokarst ponds (both polygonal ponds and
merged polygonal ponds) on NEE, we needed to eliminate the influence of polygonal ponds from our NEE estimate. To extract
uncontaminated $CO_2$ flux data from the semi-terrestrial tundra ($F_{modeled,tundra}$), we subtracted the previously estimated pond

$CO_2$ flux $F_{pond}$ from the observed $CO_2$ flux $F_{obs,mix}$:

$$F_{modeled,tundra} = \frac{F_{obs,mix} - a_{pond} \cdot F_{pond}}{a_{mix}} \tag{5}$$

We then used this estimated $CO_2$ flux from the semi-terrestrial tundra $F_{modeled,tundra}$ as the regressand variable for the *bulk-NEE model* to obtain a gap-filled dataset regarding $CO_2$ flux from the semi-terrestrial tundra. This gap-filling modeling of

$CO_2$-C flux had an RSME of 0.31 g m$^{-2}$ d$^{-1}$.

To evaluate the impact of thermokarst ponds on landscape $CO_2$ flux, we estimated a polygonal tundra landscape-$CO_2$ flux from the late-Holocene river terrace of Samoylov Island ($F_{landscape}$) by combining thermokarst ponds and semi-terrestrial tundra linearly:

$F_{landscape} = A_{pond} \cdot F_{pond} + A_{tundra} \cdot F_{modeled,tundra}$

where $F_{pond}$ describes the $CO_2$ emissions from the open-water areas of thermokarst ponds (Eq. 4), $F_{modeled,tundra}$ describes the modeled $CO_2$ flux from the semi-terrestrial tundra (Eq. 5), $A_{pond} = 0.07$ is the fraction of the river terrace area of Samoylov Island that is covered by thermokarst ponds (from the land-cover classification, see Sect. 2.4.1) and $A_{tundra} = 1 - 0.07$ is the fraction of other land-cover to the entire river terrace area. We did not account for larger or deeper lakes in this up-scaling

approach, as we expected different greenhouse gas emission dynamics from these lakes and there were no lakes in our footprint, and therefore not within our observation range. Thus, we scaled the above numbers to $A_{tundra} + A_{pond} = 1$, which results in $A_{pond} = 0.076$ and $A_{tundra} = 0.924$.

### 2.4.5   $CH_4$ flux partitioning

The data show that the $CH_4$ emissions from the heterogeneous landscape around the tower were less spatially uniform than the

$CO_2$ emissions. Therefore, we could not use a gap-filling model for $CH_4$ that was similar to the bulk model we used for $CO_2$, so we investigated $CH_4$ emissions in a different way. Based on preliminary results from our analysis and the aerial image of the study site, we focused on four wind sectors instead of extracting the fluxes from the land-cover types:

- *tundra*: At least half of the footprint consisted of dry tundra, and the wind direction was larger than 170°.
- *shore$_{50°}$*: Less than 40% of the footprint consisted of dry tundra and water comprised least 30% of the footprint. The

wind direction was between 30° < WD < 65°.
- *pond*: At least half of the footprint consisted of open water, and the wind direction was between 65° < WD < 110°.
- *shore$_{120°}$*: Less than 40% of the footprint consisted of dry tundra and water comprised at least 30% of the footprint. The wind direction was between 110° < WD < 130°.

### 2.4.6   $CH_4$ permutation test

To evaluate whether the differences in flux medians between the four wind sectors were significant, we applied a permutation test (Edgington and Onghena, 2007). In this test, we randomly assigned each 30-min flux to one of two groups and calculated

both groups' median and the differences between the group's medians. We conducted six tests in total, using all possible combinations of pairs with the four wind sectors. After repeating this step 10000 times, we plotted the resulting differences in medians in a histogram and performed a one-sample t-test to evaluate whether the observed difference in medians differed significantly ($p < 0.01$) from the randomly generated differences.

## 3 Results

### 3.1 Meteorological conditions

During the measurement period between July 11 and September 10, 2019, half-hourly air temperatures range from -0.5 °C to 27.6 °C with a mean temperature of 8.7 °C (Fig. A2, a). The maximum wind speed measured at the EC tower at a height of 2.25 m is 8.9 m s$^{-1}$ (Fig. A2, b). PAR reaches values of up to 1419 µmol m$^{-2}$ s$^{-1}$ with decreasing maximum values during the measurement period (Fig. A2, c). Throughout the measurement period, there are 28 cloudy days, determined by identifying days with low PAR-values (maximum values below $\sim$500 µmol m$^{-2}$ s$^{-1}$).

### 3.2 CO$_2$ fluxes

When inspecting the relation between observed CO$_2$ fluxes and wind direction (Fig. 2), we find that CO$_2$ fluxes exhibit high temporal variability between positive and negative CO$_2$ fluxes from most wind directions. In the wind sector between 60°–120°, the flux source area is dominated by the merged polygonal pond. The CO$_2$-C fluxes from this pond sector show smaller absolute variability (0.09 $_{-0.33}^{0.38}$ g m$^{-2}$ d$^{-1}$, Median $_{5\,\%\,\text{Percentile}}^{95\,\%\,\text{Percentile}}$) than the fluxes from all other wind directions ($-0.08\,_{-1.56}^{0.87}$ g m$^{-2}$ d$^{-1}$, Median $_{5\,\%\,\text{Percentile}}^{95\,\%\,\text{Percentile}}$). Additionally, we observe a lower respiration rate from the merged polygonal pond than from the semi-terrestrial tundra. Fig. 3 shows the observed night-time CO$_2$ fluxes plotted against the respective weighted footprint fraction of open water. We define nighttime as PAR<20 µmol m$^{-2}$ s$^{-1}$; we expect that there would only be respiration, no photosynthesis, during the night-times. We find that the fluxes decrease as the pond area contribution increases. Thus, the strength of CO$_2$ respiration shows a dependence on the contribution of open-water. We also find that low air temperatures are mostly associated with low respiration rates.

Another aspect of CO$_2$ flux variability stems from the diurnal cycle. We compare the diurnal cycle of the CO$_2$ fluxes from the merged polygonal pond (estimated in accordance with Eq. 4) and the semi-terrestrial tundra (Eq. 5, Fig. 4). The results show a less pronounced diurnal CO$_2$ cycle from the direction of the merged polygonal pond (blue) compared to the diurnal CO$_2$ cycle from the semi-terrestrial tundra (green). We combine all data from the merged polygonal pond ($F_{pond}$ in Eq. 4), which results in a CO$_2$-C flux of **0.13** $_{\textbf{0.00}}^{\textbf{0.24}}$ g m$^{-2}$ d$^{-1}$ (Median $_{25\,\%\,\text{Percentile}}^{75\,\%\,\text{Percentile}}$).

### 3.3 CH$_4$ fluxes

We plot the observed CH$_4$ fluxes against wind direction (Fig. 5). The results show that the CH$_4$ emissions peak at $\sim 120°$, where fluxes from one shoreline of the merged polygonal pond contribute to the observed flux (Fig. 1 d, from now on *shore*$_{120°}$). We

**CO<sub>2</sub>-C Flux (g m<sup>-2</sup> d<sup>-1</sup>)**

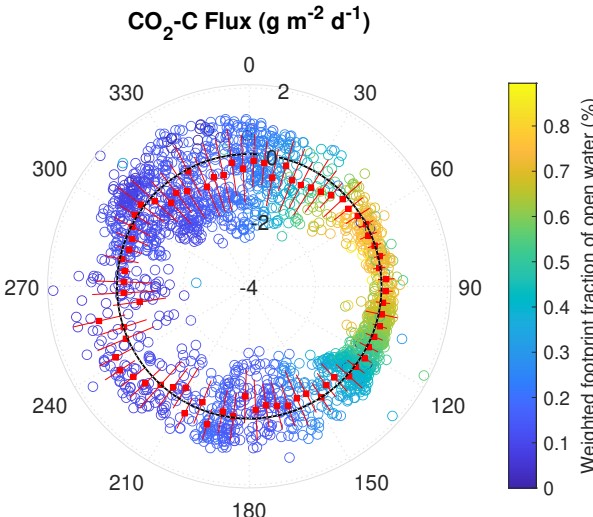

**Figure 2.** Polar plot of observed 30-minute $CO_2$-C fluxes with respect to the wind direction. Negative values (inside of the solid black line) represent $CO_2$ uptake, while positive values (outside of the dotted black line) represent $CO_2$ emissions. The values -4, -2, 0, and 2 indicate the magnitude of the $CO_2$-C flux in g m$^{-2}$ d$^{-1}$. The color of each point on the plot represents the percentage the point comprises of the total open water weighted footprint fraction in each 30-minute flux. The red boxes indicate the mean $CO_2$ flux of 5° wind direction intervals during the two-month observation period (red lines indicate the first standard deviation).

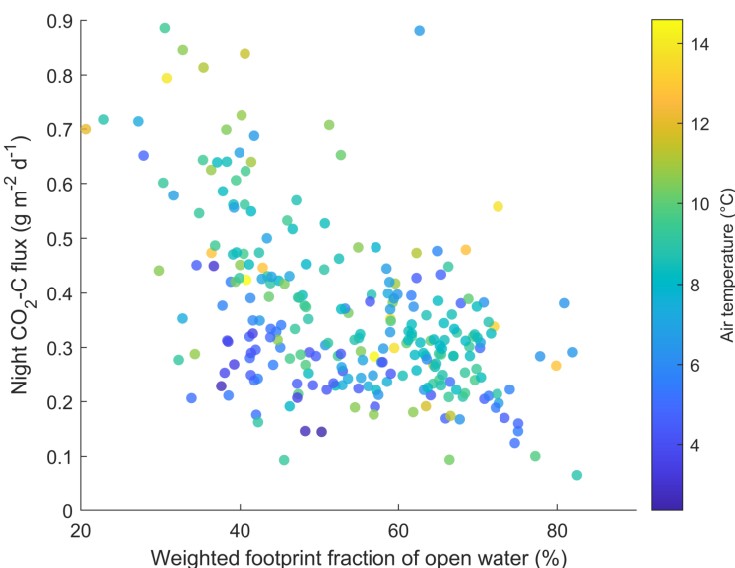

**Figure 3.** Scatter plot of observed $CO_2$ fluxes against the weighted footprint fraction of open water during each 30-minute flux. The air temperature is represented through color. Only fluxes observed at nighttime (PAR<20 µmol m$^{-2}$ s$^{-1}$) are shown.

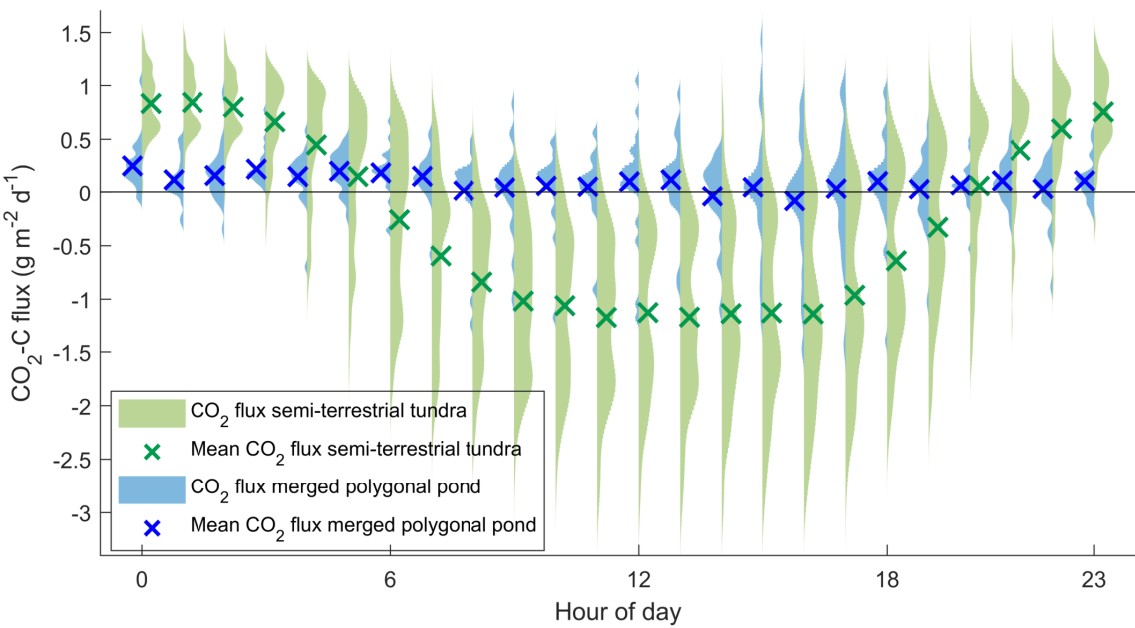

**Figure 4.** Diurnal cycle of modeled $CO_2$-C based on observations flux from the merged polygonal pond (blue, eq. 4) and the semi-terrestrial tundra (green, eq. 5) as violin plots for each half-hour flux. Blue and green crosses mark the mean $CO_2$-C flux during each half-hour flux. A violin plot shows the distribution of measurements along the y-axis – the width of the curves indicates how frequently a certain y-value occurred.

do not observe a similar peak of $CH_4$ emissions in the direction of the second shoreline towards $\sim 50°$ ($shore_{50°}$). These peaks did not correlate with a specifically large contribution of one of the land-cover classes to the footprint.

To further investigate the peak at $shore_{120°}$, we compare the $CH_4$ emissions from the different wind sectors ($shore_{120°}$, $shore_{50°}$, $pond$ and $tundra$, Sect. 2.4.5). We find the following fluxes from the wind sectors: $19.18 \, ^{24.47}_{14.26}$ mg m$^{-2}$ d$^{-1}$ ($shore_{120°}$), $12.96 \, ^{15.11}_{10.34}$ mg m$^{-2}$ d$^{-1}$ ($shore_{50°}$), $13.90 \, ^{18.46}_{11.02}$ mg m$^{-2}$ d$^{-1}$ ($pond$), and $12.55 \, ^{16.07}_{9.65}$ mg m$^{-2}$ d$^{-1}$ ($tundra$, Median $^{75\,\%\,\text{Percentile}}_{25\,\%\,\text{Percentile}}$). Fluxes from $shore_{120°}$ have a higher median than fluxes from the other three wind sectors (Fig. 6).

We investigated the impact of wind speed and air temperature on the $CH_4$ fluxes by excluding flux intervals with high wind speed (greater than 5 m s$^{-1}$) and high air temperature (warmer than 12 °C). The randomization test (Sect. 2.4.6) provided evidence of a significant difference between $CH_4$ emissions from $shore_{120°}$ and the other three wind sector classes at low wind speeds (top row in Fig. A4) and no significant difference between the $CH_4$ emission from the classes $pond$ - $tundra$ and $shore_{50°}$ - $tundra$. The difference between the classes $pond$ and $shore_{50°}$ is significant; however, it is much smaller than the previously described differences (see center graph in Fig. A4). Note that the $CH_4$ emissions from $pond$ and $tundra$ have a similar magnitude under moderate wind speed conditions. The results are very similar for moderate temperatures: We find evidence of a significant difference between the $CH_4$ emissions from $shore_{120°}$ and the $CH_4$ emissions from the other three

**Figure 5.** Polar plot of 30-minute observed $CH_4$-C flux with respect to the wind direction at the EC tower. Positive values outside the solid black line represent $CH_4$ emissions, while values and inside the line represent $CH_4$ uptake during one half-hour period. The values 0, 20, 40, and 60 indicate the magnitude of the $CH_4$-C flux in mg m$^{-2}$ d$^{-1}$. The color of each point on the plot represents the percentage the point comprises of the total open water weighted footprint fraction in each 30-minute flux. The red boxes indicate the mean $CH_4$ flux of $5°$ wind direction intervals during the two-months observation period (red lines indicate the first standard deviation).

wind sector classes (top row in Fig. A5). The differences in medians between the *pond* and *shore*$_{50°}$ and between the *pond* and *tundra* are significant. However, this difference is much smaller (second row in Fig. A5). In summary, neither high wind speed nor high temperatures act as a driver for the high $CH_4$ emission from *shore*$_{120°}$. In contrast, the peak at $180° - -190°$ can be explained reasonably well using air temperature and friction velocity in a multiple linear regression ($R^2 = 0.44$). Using the same predictors results in an $R^2$ of 0.20 for the peak at *shore*$_{120°}$.

The ratio of $CO_2$-C to $CH_4$-C emissions at night (PAR<20 μmol m$^{-2}$ s$^{-1}$) has a value of $CH_4/CO_2 = 0.060_{0.049}^{0.076}$ for fluxes with an open-water weighted footprint fraction of more than 60 %, whereas the ratio amounts to $CH_4/CO_2 = 0.020_{0.015}^{0.024}$ (Median $_{25\,\%\,\text{Percentile}}^{75\,\%\,\text{Percentile}}$) for fluxes with an open-water weighted footprint fraction of less than 20 %.

## 3.4 Upscaled $CO_2$ flux

We use the estimated open-water $CO_2$ flux from the merged polygonal pond and the modeled $CO_2$ flux from the semi-terrestrial tundra to linearly up-scale the $CO_2$ flux for the polygonal tundra of Samoylov Island (excluding larger lakes, the method described in Sect. 2.4.4). As we have not obtained estimates for the $CH_4$ fluxes from *tundra* and *pond* land-cover types, we only upscale $CO_2$.

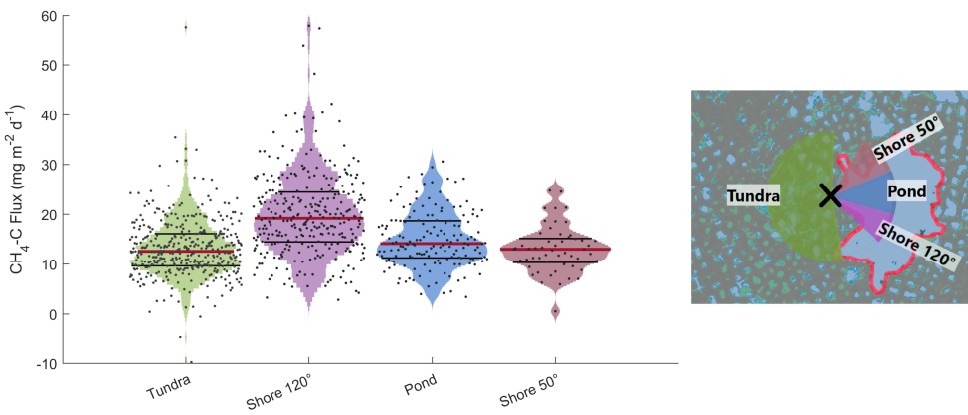

**Figure 6.** Violin plots of observed $CH_4$ emissions at the EC tower separated into four different wind sector classes. A violin plot shows the distribution of measurements along the y-axis - the width of the curves indicates how frequently a certain y-value occurred. Medians of $CH_4$ emission distributions are shown as red lines, and 75th and 25th percentile are shown as black lines. On the right, the wind sectors with the eddy covariance tower in the center (black cross) are shown.

We estimate that when one includes the $CO_2$ flux from thermokarst ponds, the river terrace landscape's $CO_2$ uptake is $\sim$ 11% lower than the uptake of semi-terrestrial tundra without ponds. The modeled $CO_2$-C flux from the semi-terrestrial tundra (without consideration of thermokarst pond fluxes) accumulated to -16.29 $\pm$ 0.43 g m$^{-2}$ during the observation period (60.5 days). If separated into months, the modeled $CO_2$-C flux from the semi-terrestrial tundra amounts to -15.01 $\pm$ 0.26, -3.56 $\pm$ 0.33 and +2.35 $\pm$ 0.11 g m$^{-2}$ in July (19.8 days), August (31 days), and September (9.7 days), respectively. When one includes the $CO_2$ flux from the merged polygonal pond to represent all thermokarst ponds on Samoylov Island, the resulting estimate of the landscape $CO_2$ flux amounts to -14.47 $\pm$ 0.40 g m$^{-2}$ (60.5 days), with monthly fluxes of -13.75 $\pm$ 0.24, -2.99 $\pm$ 0.31, and +2.27 $\pm$ 0.10 g m$^{-2}$ in July (19.8 days), August (31 days), and September (9.7 days), respectively. Thus, the results show that thermokarst ponds have the largest impact on the landscape's $CO_2$ flux in August. In September, accounting for thermokarst ponds leads to a 3.5 % lower estimate of landscape $CO_2$ emissions.

## 4 Discussion

### 4.1 $CO_2$ flux

Only a limited number of EC $CO_2$-flux studies from permafrost-affected ponds and lakes are available (studies with "EC" in Tab. 1). Estimates of open-water EC $CO_2$-C flux range from 0.059 g m$^{-2}$ d$^{-1}$ (Jammet et al., 2017), to 0.11 g m$^{-2}$ d$^{-1}$ (Eugster et al., 2003), to 0.22 g m$^{-2}$ d$^{-1}$ (Jonsson et al., 2008). Our estimate of 0.12 $_{0.0014}^{0.24}$ g m$^{-2}$ d$^{-1}$ is, therefore, well within the range of open-water $CO_2$-C fluxes observed with the EC method. Other studies using different methods report a wider range of open-water $CO_2$ fluxes in arctic regions. These fluxes range from a minor $CO_2$-C uptake (-0.14 g m$^{-2}$ d$^{-1}$,

**Table 1.** Daily mean water-atmosphere $CO_2$ & $CH_4$ fluxes from different study sites. TBL is the abbreviation for thin boundary layer model, EC for eddy covariance, CH for chamber measurement, MOD for modelled fluxes, STO for storage fluxes, and NEW for the method used in this study. All fluxes are given $\pm$ standard deviation, except of fluxes from this study are given as Median $^{75\% \text{ Percentile}}_{25\% \text{ Percentile}}$ of fluxes from this study.

| Study | Location | Period/Time | Study Site | Method | $CO_2$-C flux ($g\ m^{-2}\ d^{-1}$) | $CH_4$-C flux ($mg\ m^{-2}\ d^{-1}$) |
|---|---|---|---|---|---|---|
| This study | Lena Delta, Northern Siberia | 11.07.–10.09.2019 | merged polygonal pond | EC/NEW | $0.13^{0.24}_{0.00}$ | $14.10^{18.67}_{11.23}$ |
| | | | merged polygonal pond shore | EC | – | $12.96^{15.11}_{10.34} - 19.18^{24.47}_{14.26}$ |
| Abnizova et al. (2012) | Lena Delta, Northern Siberia | 01.08. – 21.09.2008 | Samoylov Pond | TBL | 1.50 – 2.20 | – |
| | | | Samoylov Lake | TBL | 1.40 – 2.10 | – |
| Jammet et al. (2017) | Northern Sweden | 2012 – 2013 | Lake Villasjön | EC | 0.059 | $13.42 \pm 1.64$ |
| Jonsson et al. (2008) | Northern Sweden | 17.06. – 15.10.2005 | Lake Merasjärvi | EC | $0.22 \pm 0.002$ | – |
| | | | | TBL | $0.30 \pm 0.01$ | |
| Eugster et al. (2003) | Alaska | 27.07 – 31.07.1995 | Toolik Lake | EC | $0.11 \pm 0.033$ | – |
| | | | | TBL | $0.13 \pm 0.003$ | |
| | | | | CH | $0.37 \pm 0.060$ | |
| Jansen et al. (2019) | Northern Sweden | Year round, 2010 – 2017 | Villasjön | CH | $0.22 \pm 0.047$ | $14.04 \pm 2.25$ |
| | | | Inre Harrsjön | | $0.25 \pm 0.05$ | $10.39 \pm 1.40$ |
| | | | Mellersta Harrsjön | | $0.73 \pm 0.067$ | $13.76 \pm 2.81$ |
| Bouchard et al. (2015) | NE Canada | July 2013 & 2014 | Bylot Island, Polygon ponds | TBL | -0.14 – 0.74 | 0.50 – 6432 |
| | | | Lakes | | -0.085 – 0.062 | 0.70 – 74.5 |
| Sepulveda-Jauregui et al. (2015) | Alaska | June – July 2011 & 2012 | 8 Lakes, Yedoma | TBL & STO | $0.60 \pm 0.58$ | $92.86 \pm 35.72$ |
| | | | 32 Lakes, Non-Yedoma | | $0.10 \pm 0.10$ | $16.80 \pm 8.61$ |
| Treat et al. (2018) | Northeast European Russia | 2006 – 2015 | Multiple Lakes | MOD | $0.028 \pm 0.00011$ | $0.84 \pm 0.0$ |
| Sieczko et al. (2020) | Northern Sweden | July – August 2017 | Lake Ljusvattentjärn | CH | – | $2.95 \pm 0.75$ |
| Ducharme-Riel et al. (2015) | North-East Canada | Summer 2008 | 15 lakes | TBL | $0.20 \pm 0.093$ | – |
| Repo et al. (2007) | Western Siberia | 03.07. – 06.09.2005 | MTlake | TBL | $0.14 \pm 0.11$ | – |
| | | | FTlake | TBL | $0.41 \pm 0.25$ | |
| | | | MTpond | TBL | $0.44 \pm 0.25$ | |
| Lundin et al. (2013) | Northern Sweden | 2009 (only ice-free season) | 27 lakes | TBL | $0.18 \pm 0.11$ | – |
| Kling et al. (1992) | Alaska | 1975 – 1989 | 25 lakes | TBL | $0.25 \pm 0.040$ | $5.16 \pm 0.96$ |

Bouchard et al. (2015)) to substantial emissions of $CO_2$-C (up to 2.2 g m$^{-2}$ d$^{-1}$, Abnizova et al. (2012)). A modeling study involving multiple lakes in Northeast European Russia found that they produce almost zero emissions (0.028 g m$^{-2}$ d$^{-1}$, Treat et al. (2018)).

Strikingly, our estimates of open-water $CO_2$ emissions are approximately 12–18 times smaller than those that have been previously reported for open-water $CO_2$ emissions at the same study site (Abnizova et al., 2012). One reason for the divergent results might be the different methods used. In Abnizova et al. (2012), the thin boundary layer model (TBL), following Liss and Slater (1974), was applied to estimate $CO_2$ emissions from $CO_2$ concentrations. However, one other study found good agreement between the EC method and the TBL (Eugster et al., 2003). In addition, in contrast to the larger merged polygonal pond we focus on, Abnizova et al. (2012) measured two polygonal ponds (they took 46 water samples in August and September 2008). These two ponds might have had exceptionally high $CO_2$ concentrations and might not be representative of polygonal ponds in our study area. If the polygonal ponds in the footprint of our EC measurements emitted $CO_2$ in the quantities suggested by Abnizova et al. (2012), we would expect to see their signal more clearly in our measurements.

Our approach of combining a footprint model with a land-cover classification to extract fluxes from different land-cover classes allows us to determine the thermokarst pond $CO_2$ flux. We report an uncertainty range in respect of the thermokarst pond $CO_2$ flux; however, identifying the full uncertainty of this flux is not possible using this approach due to the footprint analysis' unknown degree of uncertainty. Still, the results in respect of the thermokarst pond $CO_2$ flux are plausible and in the expected order of magnitude for two reasons. First, a reduced diurnal variability is observed when the merged polygonal pond influences the flux signal (Fig. 4). This reduction indicates that the respiration rate from the merged polygonal pond is lower than the respiration rate from the semi-terrestrial tundra, where ample oxygen is available in the upper soil layer. Additionally, since the thermokarst ponds have a lower vegetation density than the tundra, there is less photosynthesis. Second, when focusing on night-time fluxes, when only respiration occurs (i.e. no carbon is taken up), there is a decrease in $CO_2$ emissions with an increasing weighted footprint fraction of open water (Fig. 3); this also indicates that there was reduced decomposition in the merged polygonal pond. Overall, based on the data, the findings that thermokarst ponds have lower $CO_2$ emissions than the semi-terrestrial tundra are reasonable.

## 4.2 $CH_4$ flux

We observe large differences in $CH_4$ emissions from the four wind sectors. $CH_4$ emissions from $shore_{120°}$ are significantly higher than from $shore_{50°}$, *pond*, and *tundra* (Sect. 3.3). Notably, we tested the dependence of these higher fluxes on wind speed and air temperature. We expect high wind speeds to enhance turbulent mixing of the water column and diffusive $CH_4$ outgassing at the water-atmosphere interface. High wind speeds are also associated with pressure pumping, which potentially fosters the ebullition of $CH_4$. On the other hand, peak temperatures can lead to peak $CH_4$ production and emissions due to enhanced biological activity. However, the high emissions from $shore_{120°}$ do not coincide with either of two key meteorological conditions, high wind speeds and high temperatures, which would especially favor high emissions. Thus, the difference in methane flux dynamics between $shore_{120°}$ and $shore_{50°}$ is astounding since the shorelines share many other characteristics.

Both shorelines extend radially (in a fairly straight line) from the EC tower (Fig. 1), thus contributing similarly to the EC flux. The underwater topography does not vary significantly between the two shorelines. Meters away from the shore, both shorelines have a water depth of a few centimeters and a few decimeters (see data from Boike et al. (2015a)). As previously described in section 2.1, both shorelines are dominated by *Carex aquatilis*, and from visual inspection, we could not identify differences in shoot density. We, therefore, assume that the characteristics of the emergent vegetation do not play a major role in explaining the differences between the $CH_4$ emissions from $shore_{120°}$ and $shore_{50°}$. We also examine the evolution of the shorelines at the merged polygonal pond to check whether erosion along the shoreline could cause the high $CH_4$ emissions. We compare an image from 1965 (U.S. Geological Survey, EROS Center, 1965) with the current (2019) shoreline, yet we cannot identify signs of recent erosion. Furthermore, high-resolution aerial images of this pond from 2008 (Boike et al. (2015b), resolution $> 0.33$ m) and 2015 (Boike et al. (2015c), resolution $> 0.33$ m) show no signs of erosion. We therefore assume that past erosion is unlikely is unlikely to have been a factor that caused the high levels of $CH_4$ emissions we observed in 2019.

Local ebullition of the merged polygonal pond could lead to high $CH_4$ emissions from $shore_{120°}$. We applied the method proposed by Iwata et al. (2018) to check for signs of ebullition events. This method uses the 20 Hz raw $CH_4$ concentration data to detect short-term peaks in $CH_4$ that originate from ebullition events. However, we cannot detect ebullition events in the 20 Hz raw data.

In summary, meteorological conditions (wind speed and temperature), characteristics of emergent vegetation, coastal erosion, and intense ebullition events, are unlikely to be the main driving factors of the increased $CH_4$ emissions we observed. Another possible driver of higher $CH_4$ emissions from $shore_{120°}$ is a small but steady seep ebullition hot spot close to this shoreline (such as ebullition class *Kotenok* in Walter et al. (2006)). Seep ebullition hot spots have been reported to occur heterogeneously in clusters in Alaskan lakes (Walter Anthony and Anthony, 2013). Unfortunately, seep ebullition has not previously been reported in water bodies in our study area, so we did not include measurements targeting this process in our measurement campaign. In future studies, visual inspection of trapped $CH_4$ bubbles in the ice column during wintertime, as proposed by Vonk et al. (2015), could reveal more information about the cause of the higher $CH_4$ emissions from $shore_{120°}$, as could funnel or chamber measurements with high spatial coverage.

The results show that the merged polygonal pond emits a similar magnitude of $CH_4$ as the polygonal tundra surface under similar meteorological conditions and when excluding the high emissions from $shore_{120°}$. However, substrate availability and temperature dynamics differ substantially. Additionally, in dense soils, methane diffuses slowly enough through soil layers containing oxygen that the methane can be oxidized before reaching the surface. In contrast, methane emitted in ponds can reach the surface quickly through ebullition or plant-mediated transport in addition to diffusion. Therefore, we expect to see larger differences between the $CH_4$ emissions from the merged polygonal pond and the polygonal tundra, more akin to the differences that have been detected in a subarctic lake and fen by Jammet et al. (2017). However, we see no significant difference in the $CH_4$ emissions from the open-water areas of the merged polygonal pond and the polygonal tundra surface (Fig. 6 & A4).

Since many other thermokarst ponds in our study area are smaller than the merged polygonal pond (making them unsuitable to study using the EC method), and since smaller ponds tend to be greater emitters of methane (Holgerson and Raymond, 2016; Wik et al., 2016), our measurements might provide a lower limit of overall thermokarst pond $CH_4$ emissions.

We estimate a $CH_4$-C flux of $13.38^{15.92}_{10.55}$ mg m$^{-2}$ d$^{-1}$ (Median $^{75\,\%\,\text{Percentile}}_{25\,\%\,\text{Percentile}}$) from the merged polygonal pond and $12.96^{15.11}_{10.34}$–$19.18^{24.47}_{14.26}$ mg m$^{-2}$ d$^{-1}$ from the shores of this pond. This is higher than the fluxes measured by Jammet et al. (2017) from a sub-arctic lake (Tab. 1). The authors report a mean annual $CH_4$-C flux of $13.42 \pm 1.64$ mg m$^{-2}$ d$^{-1}$ and a mean ice-free season $CH_4$-C flux of $7.58 \pm 0.69$ mg m$^{-2}$ d$^{-1}$. A study focusing on 32 non-Yedoma thermokarst lakes in Alaska found $CH_4$-C emissions similar to our results ($16.80 \pm 8.61$ mg m$^{-2}$ d$^{-1}$, Sepulveda-Jauregui et al. (2015)). Also, a synthesis of

149 thermokarst water bodies north of $\sim 50°$ reports $CH_4$-C emissions in the same order of magnitude ($27.57 \pm 14.77$ mg m$^{-2}$ d$^{-1}$, Wik et al. (2016)). However, other recent studies have reported considerably lower $CH_4$-C emissions of $2.95 \pm 0.75$ mg m$^{-2}$ d$^{-1}$ in Northern Sweden (Sieczko et al., 2020) and, in contrast, a study finding $CH_4$-C emissions of up to 6,432 mg m$^{-2}$ d$^{-1}$ in Northeast Canada (Bouchard et al., 2015). The wide range of water-body methane emissions militates in favor of caution when generalizing our results, even for Samoylov Island, especially since the emissions within the merged polygonal

pond have been shown to be heterogeneous. Instead, after finding a hotspot in $CH_4$ emission at the pond shore, we would like to highlight that the gathering of additional measurements – for example employing funnel traps or counting bubbles in ice – will help to better constrain thermokarst pond $CH_4$ dynamics in their full complexity. Nevertheless, our measurements provide a robust lower limit of thermokarst pond $CH_4$ emissions.

## 4.3    Upscaling the $CO_2$ flux

We upscale the $CO_2$ emissions for the river terrace on Samoylov, an area for which we have access to a high-resolution land-cover classification. We find that we overestimate the carbon-dioxide uptake of the polygonal tundra by 11 % when we do not account for the thermokarst ponds' $CO_2$ emissions. A similar approach by Abnizova et al. (2012) found a potential increase of 35–62 % in the estimate of $CO_2$ emission from the Lena River Delta when including small ponds and lakes in the landscape $CO_2$ emission calculation. If we follow the upscaling approach by Abnizova et al. (2012) and consider overgrown water as

part of the thermokarst ponds, the estimate of the landscape $CO_2$ uptake would decrease by 19 %. Kuhn et al. (2018) also found water bodies in arctic regions to be an important source of carbon, which could outbalance the carbon dioxide uptake of the semi-terrestrial tundra in a future climate. In summary, our results demonstrate that open-water $CO_2$ emissions can substantially influence the summer carbon balance of the polygonal tundra. With respect to the night time emissions, we find that per gram $CO_2$-C thermokarst ponds emit 0.06 g $CH_4$-C whereas the semi-terrestrial tundra only emits 0.02 g $CH_4$-C.

This finding underlines again that, especially when considering thermokarst ponds, $CH_4$ emissions are of significant interest. Even though mean $CH_4$ emissions from the semi-terrestrial tundra and open water are of similar magnitude, we expect that the impact of thermokarst ponds on the carbon balance would be even greater when accounting for $CH_4$ due to locally high emissions.

     Our results suggest that future studies that aim to capture a representative landscape flux should pay extra attention to the

water bodies in their footprint. The $CO_2$ flux from thermokarst ponds has the opposite sign ($CO_2$ emission) as the semi-

terrestrial tundra ($CO_2$ uptake) during the observation period. Consequently, thermokarst ponds should cover about as much area in the measurement as they do in the landscape area of interest. In this way, the chances of capturing $CH_4$ hotspots, which can be investigated more closely, are also greater.

## 5 Conclusions

We find that thermokarst ponds are a carbon source. At the same time, the surrounding semi-terrestrial tundra in our study area acts as a carbon sink during the summer period (July–September), which is in agreement with prior studies (Abnizova et al., 2012; Jammet et al., 2017), despite that we observe much lower open-water $CO_2$ fluxes compared to previous work at the same study site (Abnizova et al., 2012). Using our approach to disentangle the EC fluxes from different land-cover classes, we posit that during the measurement period, we would overestimate the carbon-dioxide uptake of the polygonal tundra by 11 %

if thermokarst ponds were not accounted for. We expect lakes to have a similar effect on the carbon budget, though a smaller one, since lakes (a) cover a similar amount of surface area as the thermokarst ponds in our study site (Abnizova et al., 2012; Muster et al., 2012) and (b) are weaker emitters of greenhouse gases than ponds (Holgerson and Raymond, 2016; Wik et al., 2016).

In contrast to $CO_2$ emissions, which are spatially more homogeneous, small-scale heterogeneity in $CH_4$ emissions makes it

difficult to find drivers of $CH_4$ emissions. We cannot pinpoint the drivers behind the high emissions along parts of the coastline, which we surmise were potentially caused by seep ebullition. Thus, we cannot estimate the impact of this heterogeneity on the landscape scale and, therefore, refrain from upscaling $CH_4$ emissions. Additionally, the open-water fluxes presented in this paper originate from a single merged polygonal pond since the other polygonal ponds surrounding the EC tower are too small to extract their fluxes using the footprint method applied here. Thus, we do not account for the spatial variability of $CH_4$

emissions between thermokarst ponds, which can be substantial (Rehder et al., 2021; Wik et al., 2016). However, we note that open-water fluxes were of a similar magnitude as the polygonal tundra fluxes. Consequently, the main impact that thermokarst ponds have on the landscape $CH_4$ budget might occur through plant-mediated transport and local ebullition.

While being ill-suited for the study of smaller ponds, we underline that the EC method is appropriate for observing greenhouse-gas fluxes from thermokarst ponds as small as $0.024\,\mathrm{km}^2$. The EC method has a higher temporal resolution than the

TBL method. It does not disturb exchange processes like the chamber flux method, which eliminates the wind at the water surface. Especially when combining an EC footprint with a land-cover classification, one can distinguish between the contribution of different land-cover classes effectively and also study the fluxes from thermokarst ponds.

We conclude that thermokarst ponds contribute significantly to the landscape carbon budget. Changes in arctic hydrology and the concomitant changes in the water-body distribution in permafrost landscapes may cause these landscapes to change

from being overall carbon sinks to overall carbon sources.

*Code and data availability.* The data has been published at Pangaea (https://doi.org/10.1594/PANGAEA.937594). Code can be requested from the authors.

## Appendix A: Additional figures

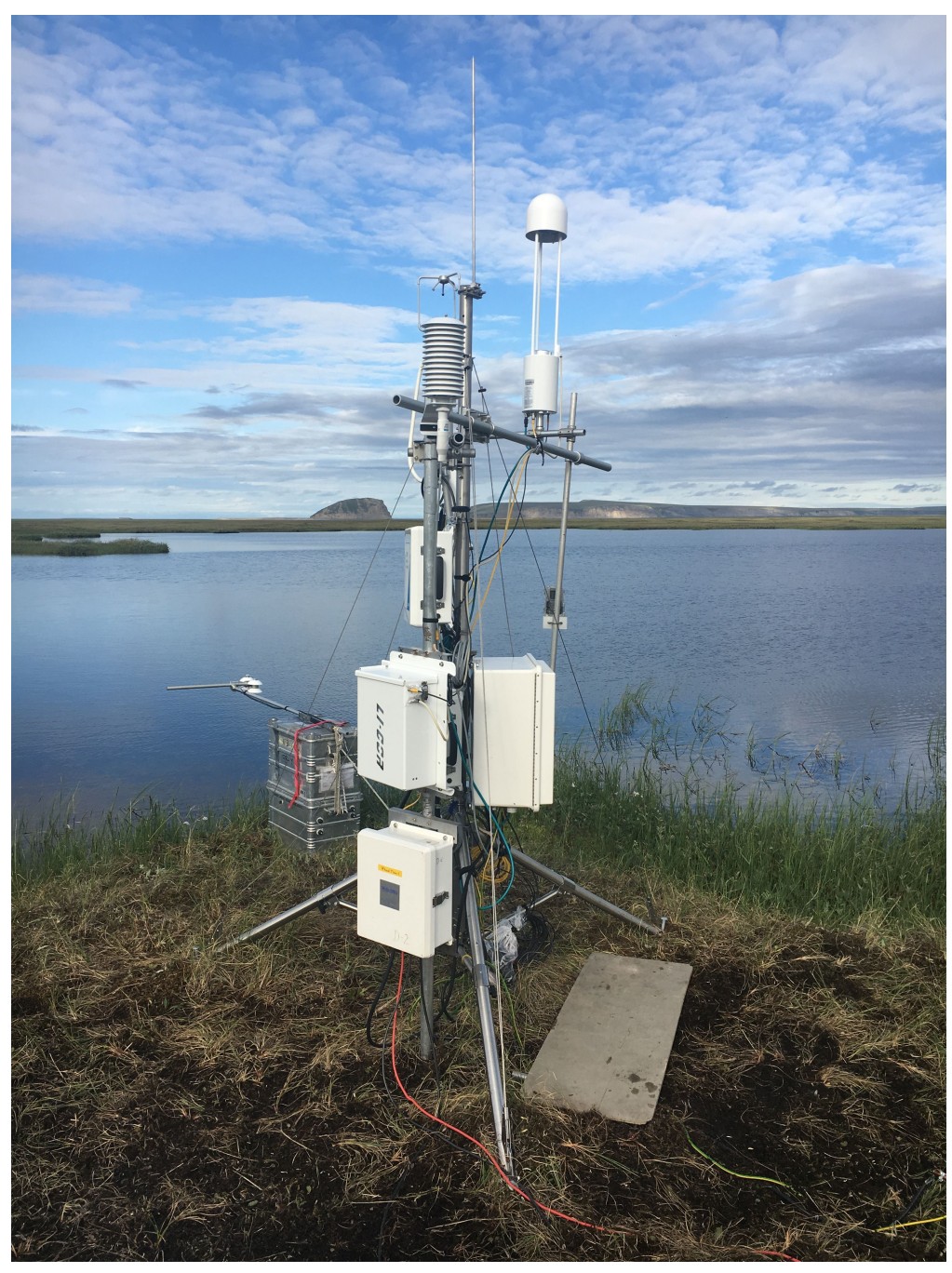

**Figure A1.** Picture of the eddy covariance tower with the merged polygonal pond in the background. Picture taken on 11 July 2019 by Zoé Rehder.

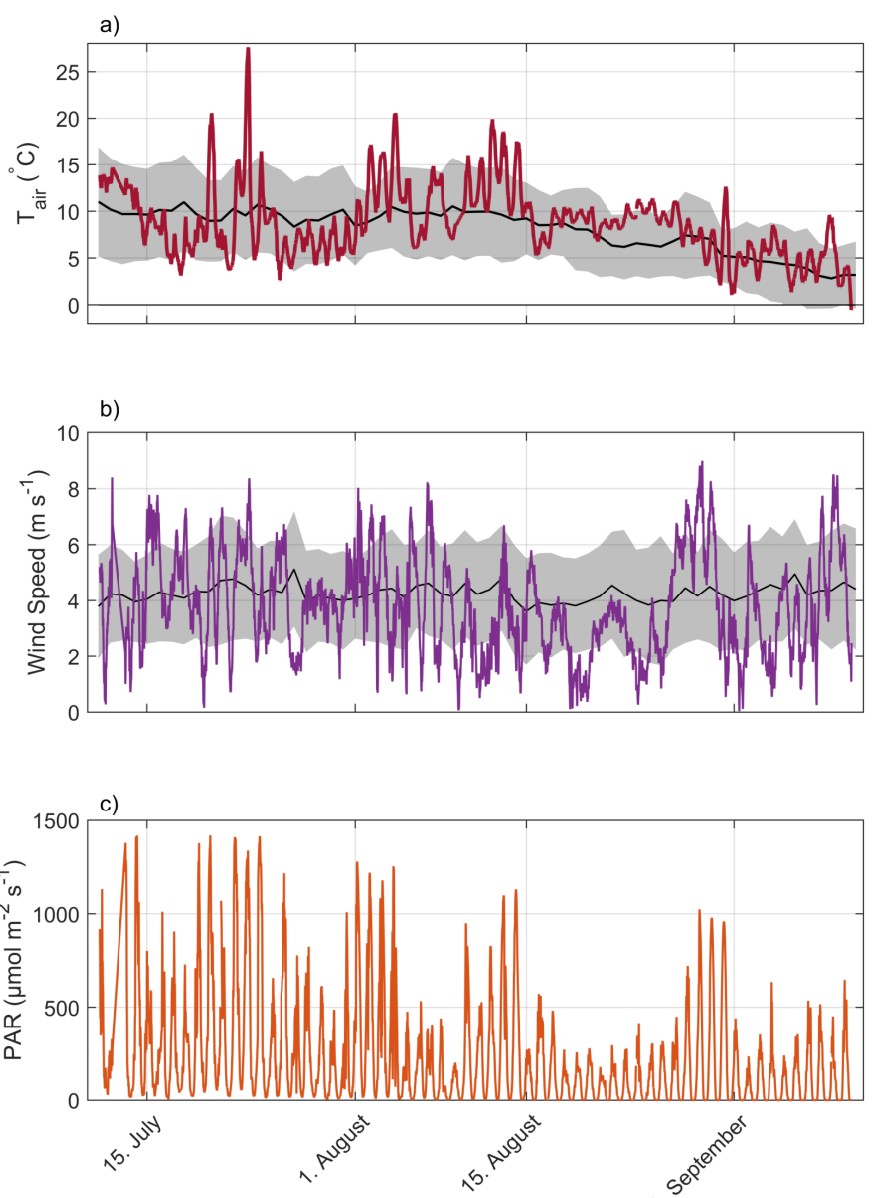

**Figure A2.** Timeline of observed meteorological conditions during the observation period with air temperature in 2 meters height (a), wind speed in 3 meters height (b) and photosynthetically active radiation (PAR) (c). Mean values and standard deviation of observations during the past 16 years are plotted as black lines and gray areas.

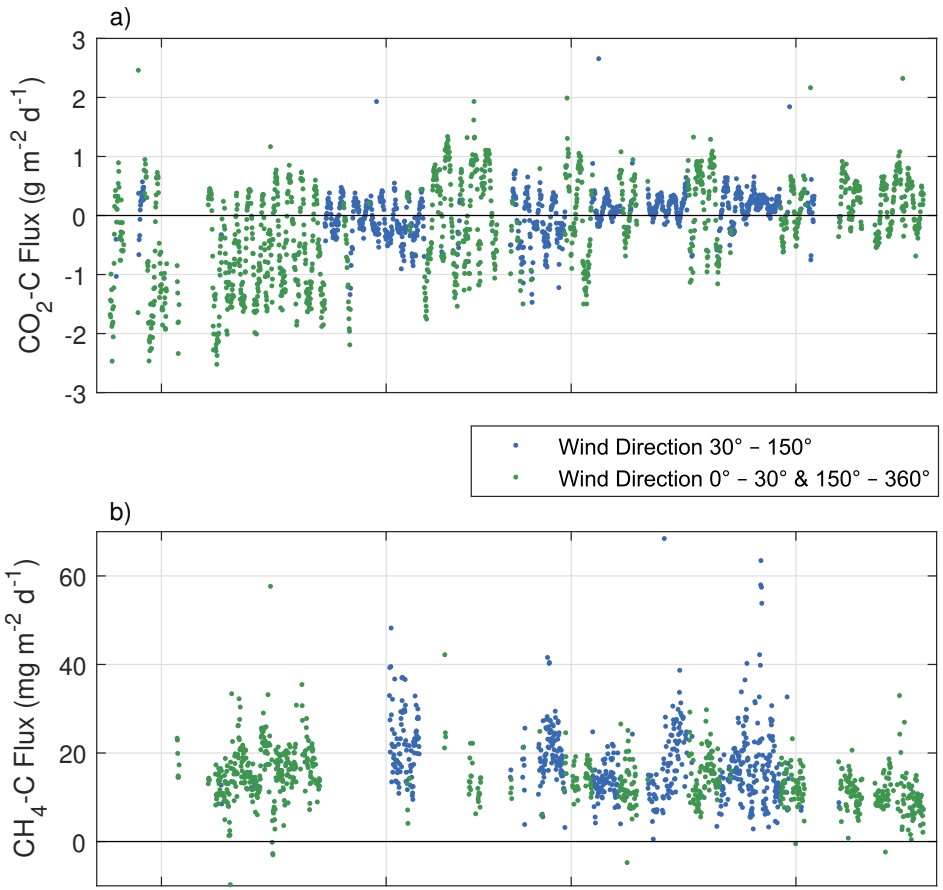

**Figure A3.** Time series of 30-minute observed $CO_2$-C flux intervals (a) and $CH_4$-C flux with a quality flag of 0 or 1. The blue color represents fluxes originating from the wind direction of the merged polygonal pond ($30° - 150°$ wind direction, mostly mixed signals from semi-terrestrial tundra and the surface of the merged polygonal pond) and the green color represents fluxes originating from all other wind directions.

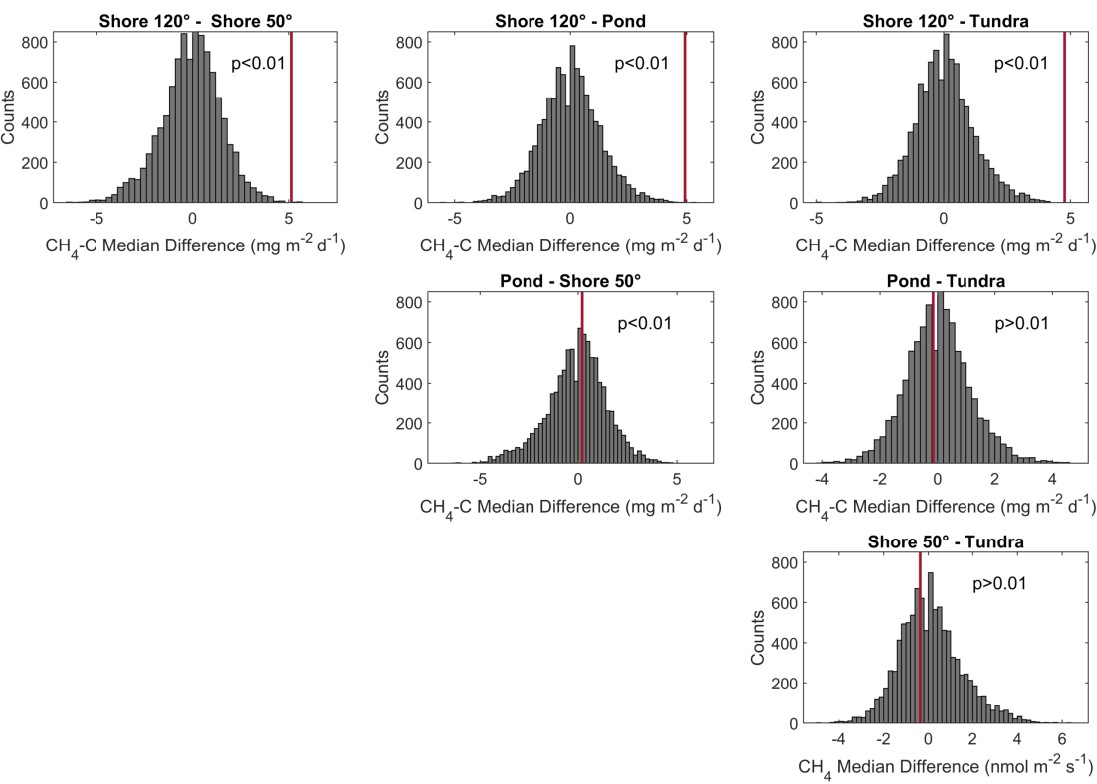

**Figure A4.** Histogram of permutation tests between the medians of CH$_4$ emissions from different wind direction classes in figure 6. All medians from flux observations during moderate wind speed conditions. The observed differences in medians between the different wind direction classes are shown in red vertical bars in each plot.

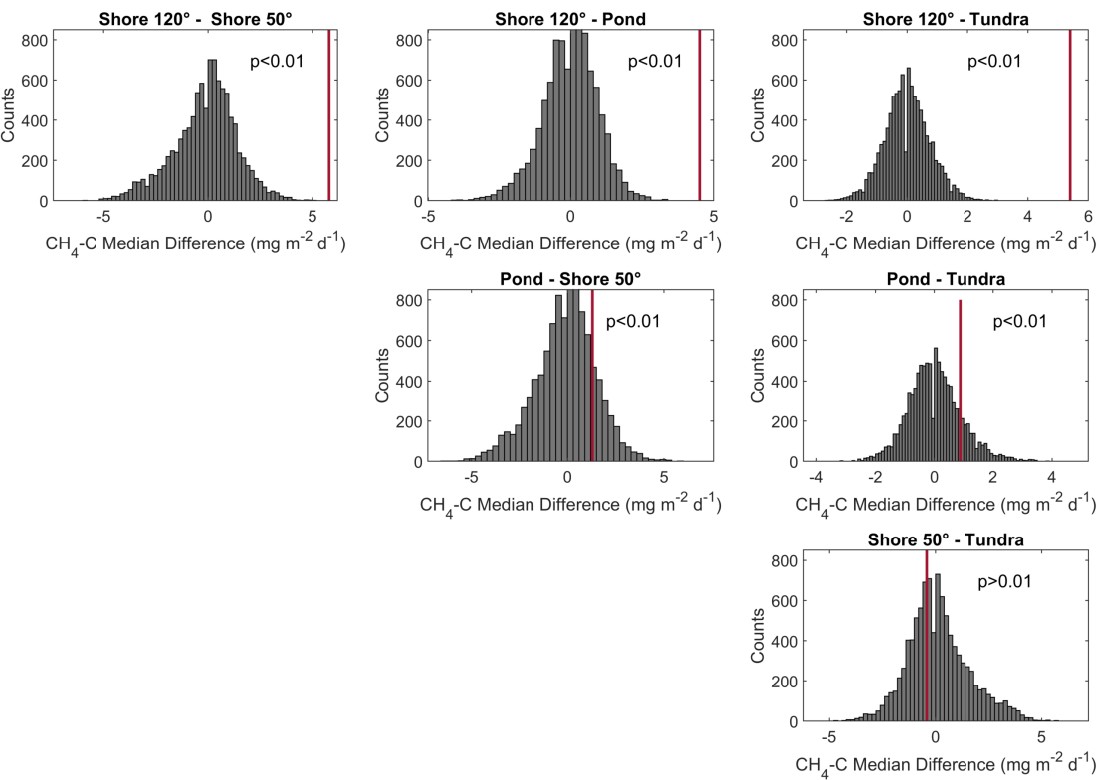

**Figure A5.** Histogram of permutation tests between the medians of CH$_4$ emissions from different wind direction classes in figure 6. All medians from flux observations during moderate air temperature conditions. The observed differences in medians between the different wind direction classes are shown in red vertical bars in each plot.

*Author contributions.* ZR and LK designed the experiments, ZR and LB carried out the fieldwork. ZR, LB, and LK developed the idea for the analysis, and CW and LB prepared the data. The formal analysis and data visualization were performed by LB and ZR with supervision by DH and LK. Resources (land-cover classification) have been provided by CM. LB and ZR prepared the manuscript with contributions from all co-authors.

*Competing interests.* The authors declare that they have no conflict of interest.

*Disclaimer.* This study was funded by the Deutsche Forschungsgemeinschaft (DFG, German Research Foundation) under Germany's Excellence Strategy – EXC 2037 'CLICCS - Climate, Climatic Change, and Society' – Project Number: 390683824, contribution to the Center for Earth System Research and Sustainability (CEN) of Universität Hamburg and by the BMBF KoPf project (grant 03F0764A).

*Acknowledgements.* The authors thank Norman Rüggen for his tireless support before and remotely during the fieldwork, Anna Zaplavnova, Andrei Astapov, and Waldemar Schneider for their equally tireless support in the field, Andrei Astapov and Katya Abramova for additional pictures in the field, Volkmar Assmann and the station crew of Samoylov Island for their logisitical support and Sarah Wiesner, Leonardo Galera, and Tim Eckahardt and for fruitful discussions during the data analysis. Also, the authors thank the reviewers.

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
