# Peer review of "Ignoring carbon emissions from thermokarst ponds results in overestimation of tundra carbon uptake"

_Biogeosciences, 2021_

## Referee Comment (RC1)

General comments

This is a relevant study that examines the role of ponds in $CO_2$ and $CH_4$ exchange of polygon tundra in Lena Delta. This research continues the long-term GHG studies conducted by the group at the Lena Delta. The results indicated that when accounting the ponds, the estimate of the tundra's net uptake of CO2 decreased by about ten percentage while the pond and tundra had equal CH4 fluxes, except that there were a CH4 flux hotspot in the pond's shoreline. Researchers focused on a pond, not the wet polygon centers, that are common but smaller landscape features. An eddy covariance measurement system was placed on the shore of a pond formed when polygons were merged. The setting allowed distinguishing and comparing fluxes from the pond and the surrounding tundra. Moreover, wind sector-based inspection showed that some of the pond's shoreline areas were sites of high CH4 fluxes. Data are, however, limited temporally and spatially, i.e. comparison of one pond vs tundra and one growing season. Unfortunately, chamber measurements and ebullition measurements to validate the hotspot's areal features were not conducted. EC methods and data analyzes are adequate. I think that the data deserve to be published, but the current manuscript version is premature. Overall, the text needs linguistic and stylistic editing, and to my opinion, the current introduction is simplistic and the descriptions of footprint partitioning and the spatial extrapolation for CO2 and CH4 are confusing and wondered why different for the two species. Discussion could be developed to give pointers what the results signify and mean for flux estimation in the area. I am sure that authors can do a rewrite that will improve the manuscript overall, incl. language, presentation of the data analyzes, and emphasizing the significance of the results in larger context. I have listed some detailed comments below.

Specific comments

Check tense throughout the text.
l. 22. About lakes and ponds: I suggest 'potential', because the pond vs lake division is not always used
l. 34. Origins of CH4: This likely varies between sites as also in the case of CO2. Sediment mineralization produces gases as well. There are earlier studies to be cited.
l. 67. closed-path or enclosed-path?

Fig 1. Could you provide a photo of the pond?

l. 127 "is shown as a gray shaded area in Figure 1, c and d" >> here and elsewhere the figure citations can be condensed. e.g. ….cumulative footprint.. (Fig. 1 c-d).

l. 132. I suggest editing the title. What's a bulk model?

l. 151. open water and no photosynthetic activity: I understand, but maybe state here that no or only small photosynthetic activity

l. 156. Would it make more sense to include also CH4 here?

l. 160. Bulk model? Is this a common expression or should it be clarified?

l.163. in a second step > second, following sentence a bit complicated

l. 181-182. I suggest combining with the previous chapter. Something wrong in the first sentence?

l. 197 > Could add the means as well.

l.210. Could you mark the shoreline sectors on the map?

l. 213-214. to the mat&methods

l. 217-221> belongs to the discussion

l. 222-226. belongs to the mat&met

l. 243. And CH4?

Fig. 6. please, explain how to interpret the violin plot in terms of distribution along the x-axis?

I would expect similar spatial analysis for CO2 and CH4. At least, both method descriptions should appear in the mat & met.

l. 267> that's a good point note. What about differences among the ponds/lakes? Vegetation or not, sediment quality, origin i.e., glacial vs thaw pond etc.

l. 303. One could carry out a measurement campaign in fine spatial resolution applying funnels and/or chambers to capture ebullition fluxes

l. 309. Maybe not to automatically expect smaller or higher emissions from one or another and one should also consider variability in amounts of available substrates and temperature among the different habitats

l. 324. you could emphasis here the fact that you found a hotspot feature and recommend spatially representative measurement and mapping. In this case, near the pond shore.

l. 330. aquatic? do you mean landscape? included lakes and ponds in the landscape balance?

---

## Referee Comment (RC2)

Small waterbodies reduce the carbon sink of a polygonal tundra landscape

Beckebanze et al.

This manuscript describes a measurement campaign conducted to measure CO2 and CH4 fluxes in a polygonal tundra in northern Siberia, with a specific interest in fluxes of a merged polygonal pond. The 2-month data has been used to show that if emissions from ponds are neglected and those from the tundra surface only are used, the upscaling to landscape fluxes underestimates the landscape CO2 uptake rate. There is not much data existing from polygonal tundra or ponds in permafrost region, and therefore I think the manuscript adds an important piece in the understanding of GHG fluxes in tundra landscape. In general, I find the paper and the data important and I suggest acceptance of the paper after minor revisions. There are certain aspects in the calculations, explanations and interpretations which are not clear to me and I think the paper could be improved by clarifying these. In addition, there are inaccuracies in the text which need to be revised before acceptance of the paper.

I suggest using overall an alternative expression for "decrease the landscape carbon sink". In my mind the sink does not decrease. I think a more descriptive expression could be e.g. "not accounting for the pond fluxes results in an overestimation of the tundra surface CO2 uptake" or similar. I understand however that in the previous literature this might have been a typical way to express the phenomenon, but it is not too late to change this.

More specific comments

- open path ch4 analyzer has been used in the study. It is well known that relatively large "Webb-corrections" are typical for that type of analyzers. I think it would be necessary to shortly discuss the implications of these corrections and perhaps mention how large they are.
- The unit of flux rate used throughout the paper is g m-2 d-1. That's OK even though not the most typically used; however it should be mentioned in the figures what do the points in the figures represent – 30 min flux, daily flux, or something else. ALso, it is crucial to tell in each figure, if just pure measurement data has been used, or if gap-filled fluxes are included also.
- I think it is always useful to see a time series of the original (screened) fluxes. Or if not, it is a good habit to tell how much data was available and if there were long gaps.

line 22: "…reduce the C sink…" consider revising (see my comment above)

line 23: impact on what?

lines 129-131: This means that there is a probability of 10% that fluxes observed at the EC tower originate from areas outside of the light gray area. Medium gray represents 50-70%, medium-dark gray 30-50%, and dark gray indicates that there is a probability of less than 30% that the observed flux originates from within the marked area.

Shouldn't it be: 10% of the flux signal originates outside the target area/fetch? Now it seems to me that you are saying that in 1/10 of the cases the whole flux data signal originates outside the area. That cannot be the case.

chapter 2.4.3: you say you use the model to partition and gap fill the NEE data. However, I do not see a mentioning of GPP or Rtot anywhere after this chapter. Also, it is not clear for me, where you have used the gapfilled data, and where you have just averaged the accepted observations. For example in Fig. 2, is this gap filled or measured data?

Line 149: "We split the datasets into a training (70%) and a validation (30%) data set to test model performance" Where do you show or discuss these test results?

chapter 2.4.4: modeled $CO_2$ flux represents the vegetated tundra. So, do you use purely modeled data here, or gap-filled? If modeled, why not gap-filled? How many gaps there are in the data?

line 160: I'm not fully convinced why do you need the fluxes from the mixed surface to conclude something about the $CO_2$ fluxes from the merged pond? Don't you have enough observations from that directions? Seeing the number of accepted observations and their distribution in time would help in understanding that.

line 161: should be >30 & < 150, right?

line 165: "Thus, we can calculate the observed $CO_2$flux…" this formulation sounds weird (why do you need to calculate the observed flux?), please consider revising, perhaps replace with "express" or something

line 170: "To improve data quality, we exclude 30-min flux intervals of Fpond when apond<50%." Now there seems to be a contradiction: in lines 162-163 you state that Fmodeled, mix includes only data with <30% of weighted footprint fraction of open water (apond?). But now you say that you exclude all Fpond values with apond < 50%. Perhaps this needs clarification.

lines 222-232: I have difficulties to follow the logic in this text. The chapter starts by stating that "To evaluate whether the differences in medians between the four wind sectors are significant, we apply a permutation test". Then fluxes are randomly assigned to one of two groups (why two? Ok, this comes evident when one looks at the appendix figures. But not from the text). What is unclear to me is that how can you conclude from the test explained here and illustrated in Appendix figures that "no meteorological parameter acted as a driver for the high $CH_4$ emission"?

Then, the $CH_4/CO_2$ ratios explained on lines 233-242: what is the conclusion from that analysis? I do not find any discussion about that.

Lines 271-272: "Our approach of combining a footprint model with a land cover classification to extract fluxes from different land cover classes allows us to determine the pond $CO_2$flux." This sentence is in the core of all my difficulties in understanding what has actually been done. Didn't you use the direct pond fluxes (from sector 60-120) to infer the pond $CO_2$ flux? At least this is what you mention on line 207, and in the table 1. And (in my understanding so far) you used the footprint model approach to estimate the flux from "tundra" (or semi-terrestrial tundra, vegetated tundra; are these same? If yes, please ease the reader's pain and use uniform expressions here. If not, please explain more clearly what's the difference.

Chapter 4.2: The observation of the $CH_4$ spike in the shore120 is interesting, and the fact that it remains unexplained, is pity but not unexceptional in flux studies! It is also somewhat convincing how much effort you have had to explore the reasons for the higher emission

line 340: a somewhat similar approach has been used also earlier, see e.g. https://bg.copernicus.org/articles/16/255/2019/

Figures 2 and 5 (which are nice and indicative figures overall!): please indicate if the fluxes consist of purely measured values, purely gap-filled, or both. If just measured values are shown, how are the mean values (in red) used in the study? If there are missing values during the day, the mean does not represent the true daily NEE. Is the mean (red value) a mean of all fluxes from that direction during the 2-month period?

Figure 3: is each dot a 30-min flux? Please explain it. "Flux intervals at night time"? Why interval, aren't these just fluxes?

Fig 4: please explain how the violin plot should be interpreted

Fig A2: what is the red line?

Figs A2 & A3 and line 310: there are no b's or c's in Fig. 6

---

## Author Comment (AC1)

**Reply to Review 1**

**General comments**

This is a relevant study that examines the role of ponds in CO2 and CH4 exchange of polygon tundra in Lena Delta. This research continues the long-term GHG studies conducted by the group at the Lena Delta. The results indicated that when accounting the ponds, the estimate of the tundra's net uptake of CO2 decreased by about ten percentage while the pond and tundra had equal CH4 fluxes, except that there were a CH4 flux hotspot in the pond's shoreline. Researchers focused on a pond, not the wet polygon centers, that are common but smaller landscape features. An eddy covariance measurement system was placed on the shore of a pond formed when polygons were merged. The setting allowed distinguishing and comparing fluxes from the pond and the surrounding tundra. Moreover, wind sector-based inspection showed that some of the pond's shoreline areas were sites of high CH4 fluxes. Data are, however, limited temporally and spatially, i.e. comparison of one pond vs tundra and one growing season. Unfortunately, chamber measurements and ebullition measurements to validate the hotspot's areal features were not conducted. EC methods and data analyzes are adequate. I think that the data deserve to be published, but the current manuscript version is premature.

Overall, the text needs linguistic and stylistic editing, and to my opinion, the current introduction is simplistic and the descriptions of footprint partitioning and the spatial extrapolation for CO2 and CH4 are confusing and wondered why different for the two species. Discussion could be developed to give pointers what the results signify and mean for flux estimation in the area.

I am sure that authors can do a rewrite that will improve the manuscript overall, incl. language, presentation of the data analyzes, and emphasizing the significance of the results in larger context. I have listed some detailed comments below.

> *Thank you very much for your thorough review. We carefully went through all the comments and hope that some of the general comments are already improved thanks to this, especially with regards to the method section. With regards to the introduction, we specified our study object better by differentiating between small water bodies and thermokarst ponds. More precisely, we propose the following addition in line 21:*
>
> The ponds in this polygonal tundra have formed almost exclusively through thermokarst processes: The ground has a high ice content, so when the ice melts, the ground subsides and thermokarst ponds form (Ellis et al., 2008). These ponds are often only as bis as one polygon, but when several polygons are inundated, larger shallow waterbodies form, which we call merged polygonal ponds (Rehder et al., 2021)
>
> *In the rest of the manuscript, we now use the term 'thermokarst ponds' instead of 'small water bodies'*

*Additionally, we propose to include the following paragraph in the discussion (line 336) to embed our study better in the grand scheme of things:*

Our results indicate that future eddy covariance flux studies aiming to capture a representative landscape flux should pay extra attention to the waterbodies in their footprint.

The $CO_2$ flux from ponds has the opposite sign to the tundra, and, consequently, ponds should cover about as much area in the footprint as they do in the larger landscape. In this way, the chances of capturing $CH_4$ hotspots are also higher, which can then be investigated in more detail. With warming in the Arctic thermokarst ponds might undergo drastic changes through permafrost degradation (Liljedahl et al., 2016). While some studies project a drainage of the landscape (Andresen and Lougheed, 2015), others anticipate enhanced formation of ponds due to thermokarst (Bring et al., 2016), at least on a regional scale. Uncertainty in the future evolution of thermokarst ponds creates a need for a better understanding of these systems.

*Lastly, we made many small linguistic and stylistic changes, which will hopefully improve the quality of the manuscript. For the sake of brevity, we will not list all changes here. However, we will of course submit a marked-up PDF with the differences to the new version when submitting a new manuscript.*

**Specific comments**

**Check tense throughout the text.**

*We now consistently use present tense.*

**l. 22. About lakes and ponds: I suggest 'potential', because the pond vs lake division is not always used**

*We hope that we understood this comment correctly. Our point here is that ponds emit more methane per area than lakes (see e.g. Holgerson and Raymond (2016); (Wik et al., 2016)). Even though many studies bin water bodies by size, and do not divide between ponds and lakes, we focus only on thermokarst ponds in our study. Ponds are most of the time much better mixed than lakes, often completely, so carbon emission from ponds show different patterns than lakes. We slightly changed the sentence in the manuscript in the following way:*

… they have a higher potential than lakes to counterbalance the carbon sink function of the surrounding tundra.

**l. 34. Origins of CH4: This likely varies between sites as also in the case of CO2. Sediment mineralization produces gases as well. There are earlier studies to be cited.**

*To expand more on methanogenesis, we added to following sentences in line 34.*

$CH_4$ is produced in the sub-aquatic soils and anoxic bottom waters (Borrel et al., 2011; Conrad, 1999; Hedderich and Whitman, 2006) . Additionally, $CH_4$ can also be produced in the oxic water column (Bogard et al., 2014; Donis et al., 2017), though this pathway only becomes significant in large waterbodies (Günthel et al., 2020) and is still under debate (Encinas Fernández et al., 2016; Peeters et al., 2019). During

methanogenesis, $CO_2$ can also formed as a byproduct (Hedderich and Whitman, 2006).

**l. 67. closed-path or enclosed-path?**

*Thank you for noticing. Of course, the Licor 7200 is an enclosed-path gas analyzer. We changed the word closed to enclosed three times in the method section.*

**Fig 1. Could you provide a photo of the pond?**

*Of course. We added a picture at the end of this document, include it in the supplements and refer to this picture in the methods.*

**l. 127 "is shown as a gray shaded area in Figure 1, c and d" >> here and elsewhere the figure citations can be condensed. e.g. ....cumulative footprint.. (Fig. 1 c-d).**

*Thank you for this suggestion, we harmonized and condensed all references to figures and tables. See example for l. 127 below.*

This sum is referred to as the cumulative footprint (gray shaded area in Fig. 1, c-d)

**l. 132. I suggest editing the title. What's a bulk model?**

*Thank you for this suggestion. We deleted the "bulk model" from the title of the section, the section has now the title:*

**Gap filling the $CO_2$ Flux**

*Additionally, we edited the first sentence of the section for clarification about the bulk model:*

To gap-fill the net-ecosystem exchange (NEE) fluxes of CO2, we use the bulk-NEE model by (Runkle et al., 2013).

**l. 151. open water and no photosynthetic activity: I understand, but maybe state here that no or only small photosynthetic activity**

*This is a good suggestion; we altered the sentence as follow:*

… we expect little to no photosynthetic activity in the open-water part of the merged polygonal pond.

**l. 156. Would it make more sense to include also CH4 here?**

*Instead of adding a paragraph here, we merge the two paragraphs 2.4.4 Aquatic $CO_2$ Flux and 2.4.5 Up-scaled $CO_2$ flux (as suggested by you below) and add another paragraph focusing on methane after line 188:*

2.4.5 CH4 flux partitioning

Since we do not have a simple gap filling model at hand for $CH_4$ emissions from the tundra, and since $CH_4$ emissions are much more variable than $CO_2$ emissions, we treat $CH_4$ differently. Instead of extracting the fluxes from the landcover types, we focus on wind sectors. We divide fluxes from

- *tundra*: At least half of the footprint consists of dry tundra, and the wind direction is larger than 170°.
- *shore$_{50°}$*: Less than 40% of the footprint consists of dry tundra and water contributed to the footprint with at least 30%. The wind direction lies between 30° and 65°.
- *pond*: At least half of the footprint consists of open water and the wind direction lies between 65° and 110.

- *shore$_{120°}$*: Less than 40% of the footprint consists of dry tundra and water contributed to the footprint with at least 30%. The wind direction lies between 110° and 130°.

**l. 160. Bulk model? Is this a common expression or should it be clarified?**

*We followed your suggestion and combined it with your comment on line 132. We now write "bulk model" in cursive to clarify that it is a name of the model given by Runkle et al. 2013 and not a common expression. We edited the name of the model throughout the text.*

**l.163. in a second step > second, following sentence a bit complicated**

*Thank you for pointing out this sentence. We tried to simplify the sentence as follows:*
Second, we assume that the total, observed flux is a linear combination of the fluxes from the land cover types weighted by their respective contribution to the footprint.

**l. 181-182. I suggest combining with the previous chapter. Something wrong in the first sentence?**

*As mentioned above, we followed your suggestion and combined the two sections. Additionally, we simplified the sentence in l. 180 as follow:*
To evaluate the impact of ponds on the landscape $CO_2$ flux, we estimate a polygonal-tundra landscape-$CO_2$ flux ($F_{Landscape}$) by linearly combining ponds and semi-terrestrial tundra…

**l. 197 > Could add the means as well.**

*Thank you for this comment. We added the CO2 flux-mean from the two sectors in front of the standard deviation.*

**l.210. Could you mark the shoreline sectors on the map?**

*The figure of the study site was already relatively full of information. Therefore, we decided to mark the shoreline sectors on a small map and added it to figure 6. Please see below the new figure 6.*

[Figure]

Figure 6. Violin plots of observed $CH_4$ emissions at the EC tower separated into four different wind direction classes. A violin plot shows the distribution of measurements along the y-axis - the width of the curves expresses the density of data points at each y-value. Medians of $CH_4$ emission distributions are shown as red lines, and 75th & 25th percentile are shown as black lines. On the right, the wind sectors with the eddy covariance tower in the center (black cross) are shown.

**l. 213-214. to the mat&methods**

*Since we added the paragraph on $CH_4$ after line 188, we adapt l. 213-214 as follows:*

To further investigate the peak at $shore_{120°}$, we compare the $CH_4$ emissions from the different wind sectors...

**l. 217-221> belongs to the discussion**

*We slightly altered the paragraph (see below) to fit into the discussion section and include it in line 283.*

Notably, we tested a dependence of these higher fluxes on wind speed and temperature. We expect high wind speeds to enhance turbulent mixing of the water column and diffusive $CH_4$ outgassing at the water-atmosphere interface. High wind speeds are also associated with pressure pumping, which potentially fosters ebullition of $CH_4$. Peak temperatures, on the other hand, can lead to peak $CH_4$ production and emissions due to enhanced biological activity. However, the high emissions from $shore_{120°}$ do not coincide with meteorological conditions which would especially favor high emissions.

**l. 222-226. belongs to the mat&met**

*Thank you for this helpful comment. We moved the paragraph to the new method subsection "$CH_4$ Permutation Test" and referred to this paragraph in the result section.*

**l. 243. And CH4?**

*Due to the strong variability of the $CH_4$ fluxes and the lack of a well-tested model for tundra or water $CH_4$ fluxes, we could not extract the fluxes from the individual landcover types for $CH_4$. This extraction is a prerequisite for the upscaling. Thus, the upscaling was only conducted for $CO_2$. To highlight this at this point in the manuscript we added the following sentence after line 246:*

As we have no estimates for the $CH_4$ fluxes from the landcover types *tundra* and *pond*, we only upscale $CO_2$.

**Fig. 6. please, explain how to interpret the violin plot in terms of distribution along the x-axis?**

*Thank you for this suggestion. We added to following text at the end of Fig. 4 and 6:*

A violin plot shows the distribution of measurements along the y-axis - the width of the curves expresses the density of data points at each y-value.

**I would expect similar spatial analysis for CO2 and CH4. At least, both method descriptions should appear in the mat & met.**

*By adding two paragraphs on $CH_4$ in the method section (CH_4 Flux Partitioning and CH_4 Permutation Test) and by clarifying why we do not upscale the $CO_2$ fluxes, we hope that this comment is already covered. We would like to stress here that the spatial and temporal patterns of the two gasses are so different that we do not think a similar analysis would give meaningful results. We hope that adding the method descriptions is an acceptable alternative to the reviewer.*

**l. 267> that's a good point note. What about differences among the ponds/lakes? Vegetation or not, sediment quality, origin i.e., glacial vs thaw pond etc.**

*Thank you for this helpful remark. Yes, Abnizova et al. (2012) did not measure the merged polygonal pond which we focus on, we extend the discussion focusing on the pond characteristics as follow in line 270:*

Abnizova et al. (2012) measured smaller thermokarst ponds, as opposed to the larger merged polygonal pond we focus on. While this might explain the deviations, there are also thermokarst ponds highly similar to the ones in Abnizova et al. (2012) in the footprint of the EC tower in this study. If those ponds would emit CO$_2$ in the quantities suggested by Abnizova et al. (2012), we would expect to see their signal more clearly in our measurements.

**l. 303. One could carry out a measurement campaign in fine spatial resolution applying funnels and/or chambers to capture ebullition fluxes**

*Thank you for the suggestion, we added these possibilities. Please see below for the adjusted text.*

So, a future visual inspection of trapped CH$_4$ bubbles in the ice column during wintertime, as proposed in Vonk et al. (2015), could reveal more information about the cause of the higher CH$_4$ emission from *shore$_{120°}$*, as could funnel or chamber measurements with high spatial coverage.

**l. 309. Maybe not to automatically expect smaller or higher emissions from one or another and one should also consider variability in amounts of available substrates and temperature among the different habitats**

*Thank you for this suggestion. We clarify the reasons for the different expectations in the following way.*

*Considering the first part of your suggestion, we now base our expectations additionally on a study from Abisko, Sweden. We changed the text as follows:*

However, substrate availability and temperature dynamics differ substantially. Additionally, in dense soils, methane diffuses through upper soil layers and can oxidize before reaching the surface. In contrast, methane emitted in ponds can reach the surface quickly through ebullition or higher plant-mediated transport in addition to diffusion. Therefore, we expect bigger differences between CH$_4$ emissions from the pond and the tundra, more similar to the differences detected in a subarctic lake and fen (Jammet et al., 2017).

**l. 324. you could emphasis here the fact that you found a hotspot feature and recommend spatially representative measurement and mapping. In this case, near the pond shore.**

*Thank you for this suggestion. We edited the sentences as follows:*

Instead, after finding a hotspot in CH$_4$ emission at the pond shore, we would like to highlight the need for spatially representative observation and mapping of CH$_4$ fluxes to better understand the variability of pond-CH$_4$ emissions.

**l. 330. aquatic? do you mean landscape? included lakes and ponds in the landscape balance?**

*Thank you for this comment. After reading Abnizova et al. (2012) again, we replaced "aquatic" with "landscape" at the end of the sentence and hope that we cite the paper correctly by saying:*

A similar approach by Abnizova et al. (2012) found a potential increase of 35 - 62 % in the estimate of $CO_2$ emission from the Lena River Delta when including small ponds and lakes into the landscape $CO_2$ emission.

[Figure]

Figure A1. Picture of the eddy covariance tower with the merged polygonal pond in the background. Picture taken on 11 July 2019 by Zoé Rehder.

**Literature**

Abnizova, A., Siemens, J., Langer, M. and Boike, J., 2012. Small ponds with major impact: The relevance of ponds and lakes in permafrost landscapes to carbon dioxide emissions. Global Biogeochemical Cycles, 26.

Andresen, C.G. and Lougheed, V.L., 2015. Disappearing Arctic tundra ponds: Fine-scale analysis of surface hydrology in drained thaw lake basins over a 65year period (1948-2013). Journal of Geophysical Research-Biogeosciences, 120(3): 466-479.

Bogard, M.J., del Giorgio, P.A., Boutet, L., Chaves, M.C.G., Prairie, Y.T., Merante, A. and Derry, A.M., 2014. Oxic water column methanogenesis as a major component of aquatic CH4 fluxes. Nature Communications, 5(1): 5350.

Borrel, G., Jézéquel, D., Biderre-Petit, C., Morel-Desrosiers, N., Morel, J.-P., Peyret, P., Fonty, G. and Lehours, A.-C., 2011. Production and consumption of methane in freshwater lake ecosystems. Research in Microbiology, 162(9): 832-847.

Bring, A., Fedorova, I., Dibike, Y., Hinzman, L., Mard, J., Mernild, S.H., Prowse, T., Semenova, O., Stuefer, S.L. and Woo, M.K., 2016. Arctic terrestrial hydrology: A synthesis of processes, regional effects, and research challenges. Journal of Geophysical Research-Biogeosciences, 121(3): 621-649.

Conrad, R., 1999. Contribution of hydrogen to methane production and control of hydrogen concentrations in methanogenic soils and sediments. FEMS Microbiology Ecology, 28(3): 193-202.

Donis, D., Flury, S., Stöckli, A., Spangenberg, J.E., Vachon, D. and McGinnis, D.F., 2017. Full-scale evaluation of methane production under oxic conditions in a mesotrophic lake. Nature Communications, 8(1): 1661.

Ellis, C.J., Rochefort, L., Gauthier, G. and Pienitz, R., 2008. Paleoecological Evidence for Transitions between Contrasting Landforms in a Polygon-Patterned High Arctic Wetland. Arctic, Antarctic, and Alpine Research, 40(4): 624-637.

Encinas Fernández, J., Peeters, F. and Hofmann, H., 2016. On the methane paradox: Transport from shallow water zones rather than in situ methanogenesis is the major source of CH4 in the open surface water of lakes. Journal of Geophysical Research: Biogeosciences, 121(10): 2717-2726.

Günthel, M., Klawonn, I., Woodhouse, J., Bižić, M., Ionescu, D., Ganzert, L., Kümmel, S., Nijenhuis, I., Zoccarato, L., Grossart, H.-P. and Tang, K.W., 2020. Photosynthesis-driven methane production in oxic lake water as an important contributor to methane emission. Limnology and Oceanography, 65(12): 2853-2865.

Hedderich, R. and Whitman, W.B., 2006. Physiology and Biochemistry of the Methane-Producing Archaea. In: M. Dworkin, S. Falkow, E. Rosenberg, K.-H. Schleifer and E. Stackebrandt (Editors), The Prokaryotes: Volume 2: Ecophysiology and Biochemistry. Springer New York, New York, NY, pp. 1050-1079.

Holgerson, M.A. and Raymond, P.A., 2016. Large contribution to inland water CO2 and CH4 emissions from very small ponds. Nature Geoscience, 9(3): 222-U150.

Jammet, M., Dengel, S., Kettner, E., Parmentier, F.J.W., Wik, M., Crill, P. and Friborg, T., 2017. Year-round CH4 and CO2 flux dynamics in two contrasting freshwater ecosystems of the subarctic. Biogeosciences, 14(22): 5189-5216.

Liljedahl, A.K., Boike, J., Daanen, R.P., Fedorov, A.N., Frost, G.V., Grosse, G., Hinzman, L.D., Iijma, Y., Jorgenson, J.C., Matveyeva, N., Necsoiu, M., Raynolds, M.K., Romanovsky, V.E., Schulla, J., Tape, K.D., Walker, D.A., Wilson, C.J., Yabuki, H. and Zona, D., 2016. Pan-Arctic ice-wedge degradation in warming permafrost and its influence on tundra hydrology. Nature Geoscience, 9(4): 312-318.

Peeters, F., Encinas Fernandez, J. and Hofmann, H., 2019. Sediment fluxes rather than oxic methanogenesis explain diffusive CH4 emissions from lakes and reservoirs. Scientific Reports, 9(1): 243.

Rehder, Z., Zaplavnova, A. and Kutzbach, L., 2021. Identifying Drivers Behind Spatial Variability of Methane Concentrations in East Siberian Ponds. Frontiers in Earth Science, 9(183).

Runkle, B.R.K., Sachs, T., Wille, C., Pfeiffer, E.M. and Kutzbach, L., 2013. Bulk partitioning the growing season net ecosystem exchange of $CO_2$ in Siberian tundra reveals the seasonality of its carbon sequestration strength. Biogeosciences, 10(3): 1337-1349.

Wik, M., Varner, R.K., Anthony, K.W., MacIntyre, S. and Bastviken, D., 2016. Climate-sensitive northern lakes and ponds are critical components of methane release. Nature Geoscience, 9(2): 99-+.

---

## Author Comment (AC2)

**Reply to Review 2**

**General comments**

This manuscript describes a measurement campaign conducted to measure CO2 and CH4 fluxes in a polygonal tundra in northern Siberia, with a specific interest in fluxes of a merged polygonal pond. The 2-month data has been used to show that if emissions from ponds are neglected and those from the tundra surface only are used, the upscaling to landscape fluxes underestimates the landscape CO2 uptake rate. There is not much data existing from polygonal tundra or ponds in permafrost region, and therefore I think the manuscript adds an important piece in the understanding of GHG fluxes in tundra landscape. In general, I find the paper and the data important and I suggest acceptance of the paper after minor revisions. There are certain aspects in the calculations, explanations and interpretations which are not clear to me and I think the paper could be improved by clarifying these. In addition, there are inaccuracies in the text which need to be revised before acceptance of the paper.

I suggest using overall an alternative expression for "decrease the landscape carbon sink". In my mind the sink does not decrease. I think a more descriptive expression could be e.g. "not accounting for the pond fluxes results in an overestimation of the tundra surface CO2 uptake" or similar. I understand however that in the previous literature this might have been a typical way to express the phenomenon, but it is not too late to change this.

> *Thank you for your kind words and thorough review. Please find our point-by-point replies below. Regarding the „decrease the landscape carbon sink", we agree that a different phrase might be more appropriate and changed all relevant places in the manuscript along the lines of ‚not accounting for pond emissions leads to an overestimation". This also includes the title. We propose a new title:*
> **Not accounting for thermokarst ponds leads to overestimation of tundra carbon uptake**

**More specific comments**

**open path ch4 analyzer has been used in the study. It is well known that relatively large "Webb-corrections" are typical for that type of analyzers. I think it would be necessary to shortly discuss the implications of these corrections and perhaps mention how large they are.**

> *Thank you for these suggestions. We added a discussion of these corrections in the methods:*
> This correction depends on accurate measurements of the latent and sensible heat flux and is applied to the open-path data of the LI-7700. Especially for the LI-7700, the correction term can be larger than the flux itself, but the correction is derived from the underlying physical equations. By using *EddyPro,* which uses an up-to-date implementation of the correction, and by using well calibrated instruments, we are certain to receive accurate $CH_4$ flux estimations from the LI-7700.

**The unit of flux rate used throughout the paper is g m-2 d-1. That's OK even though not the most typically used; however it should be mentioned in the figures what do the points in the figures represent – 30 min flux, daily flux, or something else. ALso, it is crucial to tell in each figure, if just pure measurement data has been used, or if gap-filled fluxes are included also.**

*This is a very valid point. We changed this in all plots. One example:*
Polar plot of 30-minute observed $CO_2$-C flux with respect to the wind direction at the EC tower.

**I think it is always useful to see a time series of the original (screened) fluxes. Or if not, it is a good habit to tell how much data was available and if there were long gaps.**

*We added a new figure to the supplement with a time series of $CO_2$ and $CH_4$ flux data.*

[Figure]

Figure A3. Time series of 30-minute observed $CO_2$-C flux intervals (a) and $CH_4$-C flux with a quality flag of 0 or 1. The blue color represents fluxes originating from the wind direction of the lake (30°–150° wind direction, mostly mixed signals from semi-terrestrial tundra and the lake surface) and the green color represents fluxes originating from all other wind directions.

**line 22: "...reduce the C sink..." consider revising (see my comment above)**

*We agree to revise this. Please refer to our answer above to the general comments.*

**line 23: impact on what?**

*Thank you for this comment. We added "on the landscape-scale carbon flux" in this sentence.*

**lines 129-131: This means that there is a probability of 10% that fluxes observed at the EC tower originate from areas outside of the light gray area. Medium gray represents 50-70%, medium-dark gray 30-50%, and dark gray indicates that there is a probability of less than 30% that the observed flux originates from within the marked area.**
**Shouldn't it be: 10% of the flux signal originates outside the target area/fetch? Now it seems to me that you are saying that in 1/10 of the cases the whole flux data signal originates outside the area.**
**That cannot be the case.**

> *That is a good suggestion. We changed the wording in the method section:*
> This means that it is likely that 10% of each observed flux signal originates from outside of the light gray area. Medium gray represents 50-70%, medium-dark gray 30-50%, and dark gray indicates that there is a probability of less than 30% that each observed flux signal originates from within the marked area.

**chapter 2.4.3: you say you use the model to partition and gap fill the NEE data. However, I do not see a mentioning of GPP or Rtot anywhere after this chapter. Also, it is not clear for me, where you have used the gapfilled data, and where you have just averaged the accepted observations. For example in Fig. 2, is this gap filled or measured data?**

> *We changed the first paragraph as follows:*
> To gap fill the net-ecosystem exchange (NEE) fluxes of $CO_2$ we use the *bulk-NEE model* by Runkle et al. (2013). The model uses the total ecosystem respiration (TER) and the gross primary production (GPP) to gap fill NEE, our target variable.

> *TER and GPP are needed for gap-filling, we do not study these quantities themselves. Additionally, we add information each time we introduce a new measurement variable to explain if this is pure observation, gap filled, or a pure model result.*

**Line 149: "We split the datasets into a training (70%) and a validation (30%) data set to test model performance" Where do you show or discuss these test results?**

> *Thank you for pointing this out. We added two sentences in this paragraph discussing the model performance briefly:*
> In 38 5-day fitting periods, we evaluate an $R^2$ above 0.9 between the model output and the validation set, 18 times an $R^2$ between 0.8 - 0.9 and six times an $R^2$ below 0.7. This indicates that the model works well overall.

**chapter 2.4.4: modeled CO2 flux represents the vegetated tundra. So, do you use purely modeled data here, or gap-filled? If modeled, why not gap-filled? How many gaps there are in the data?**

> *We added details to clarify the usage of modelled and gap-filled $CO_2$ flux, please see below.*
> To estimate the $CO_2$ flux from the merged polygonal pond ($F_{pond}$), we first fit the *bulk model* to data excluding fluxes from the merged polygonal pond (thus exclude fluxes >30° & <150° wind direction, as described in section 2.4.3).  With this bulk model, we gap fill the $CO_2$ flux, and the gap-filled $CO_2$ flux ($F_{modeled,mix}$) represents the semiterrestrial tundra surrounding the EC tower, including small ponds to the north, west and south.

**line 160: I'm not fully convinced why do you need the fluxes from the mixed surface to conclude something about the CO2 fluxes from the merged pond? Don't you have enough observations from that directions? Seeing the number of accepted observations and their distribution in time would help in understanding that.**

*We use this, admittedly slightly round-about method to extract the pond fluxes, because the merged polygonal pond is still fairly small compared to the Eddy-footprint. From Fig. 1 we learn that sometimes as much as 10% of the observed EC measurements originate from the tundra behind the merged polygonal pond, thus we have a mixed signal from all directions, also from the direction of the pond. This is why we use the bulk model to extract the pond $CO_2$ fluxes.*

*Since you already suggested that we show the accepted observations (without gap-filling), we added information about the wind sectors to these fluxes. This new figure is appended to this document and also visible in the answer of the third comment. Finally, we tried to clarify the above argumentation by rewriting the paragraph:*

Due to the strong heterogeneity of the landscape and the relatively small size of the merged polygonal pond compared to the EC footprint we measure a mixed signal from all wind directions. In other words, each flux measured with the EC method contains information from different land cover types. However, we want to extract fluxes from ponds and semi-terrestrial tundra to analyze the influence of ponds on a polygonal tundra landscape. Since we are interested in average tundra fluxes, we combine the landcover classes dry tundra, wet tundra, and overgrown water under the term *semi-terrestrial tundra*. In this way we can compare two landcover classes, semi-terrestrial tundra and the open water from thermokarst ponds.

**line 161: should be >30 & < 150, right?**

*Yes, correct, we changed this.*

**line 165: "Thus, we can calculate the observed CO2flux..." this formulation sounds weird (why do you need to calculate the observed flux?), please consider revising, perhaps replace with "express" or something**

*Thank you for this helpful suggestion. We improved the writing as followed:*

Thus, we postulate that the observed CO2 flux ($F_{obs,mix}$, not gap-filled) is the sum of the individual land cover type fluxes ($F_{modeled,mix}$ and $F_{pond}$) each multiplied with their weighted footprint fraction ($a_{tundra}$ and $a_{pond}$), ...

**line 170: "To improve data quality, we exclude 30-min flux intervals of Fpond when apond<50%." Now there seems to be a contradiction: in lines 162-163 you state that Fmodeled, mix includes only data with <30% of weighted footprint fraction of open water (apond?). But now you say that you exclude all Fpond values with apond < 50%. Perhaps this needs clarification.**

*Thank you for this helpful comment. We see ourselves that this is contradictory. The sentence in brackets in line 162-163 was added after a suggestion of the co-authors without re-evaluation the effect on the following text, and 50% is the correct number. The text in brackets in line 162-163 does not necessarily improve the understanding of*

*the text. Therefore, we clarified that $F_{modelled,mix}$ is modelled and gap-filled and deleted the text in the brackets. The paragraph around line 162 now reads as follow:*

To estimate the $CO_2$ flux from the merged polygonal pond ($F_{pond}$), we first fit the *bulk model* to data excluding fluxes from the direction of the merged polygonal pond (thus exclude fluxes >30° & <150° wind direction, as described in section 2.4.3). With this bulk model, we gap fill the $CO_2$ flux, and the gap-filled $CO_2$ flux ($F_{modeled,mix}$) represents the semi-terrestrial tundra surrounding the EC tower including small ponds to the north, west and south.

**lines 222-232: I have difficulties to follow the logic in this text. The chapter starts by stating that "To evaluate whether the differences in medians between the four wind sectors are significant, we apply a permutation test". Then fluxes are randomly assigned to one of two groups (why two? Ok, this comes evident when one looks at the appendix figures. But not from the text).**
**What is unclear to me is that how can you conclude from the test explained here and illustrated in Appendix figures that "no meteorological parameter acted as a driver for the high CH4 emission"?**

*Thank you for this comment. We can understand the difficulties to follow the logic here. From prior studies in the study area, we expected that of all meteorological variables high wind speed or temperature would be most likely to impact the methane emissions. So, we performed the permutation test using these two variables. To clarify this, we changed* "we find no meteorological parameters acting as a driver for the high $CH_4$ emission" *to* "we find neither high wind speed nor high temperature acting as a driver for the high $CH_4$ emission"
*For other available meteorological parameters, like air pressure, we tested for correlation with methane emissions, with no success.*

**Then, the CH4/CO2 ratios explained on lines 233-242: what is the conclusion from that analysis? I do not find any discussion about that.**

*You are right. This was again caused by a small misunderstanding with a coauthor. We shortened the paragraph in the results section as follow:*

The ratio of $CO_2C$ to $CH_4$ emissions at night (PAR < 20 μmol m$^{-2}$ s$^{-1}$) has a value of $CH_4/CO_2= 0.060_{0.049}^{0.076}$ when including fluxes with an open-water weighted footprint fraction of more than 60\%, whereas the ratio amounts to ($CH_4/CO_2=0.020_{0.015}^{0.024}$, $Median_{25\% \; Percentile}^{75\% \; Percentile}$ when including fluxes with an open-water weighted footprint fraction of less than 20%.

*These results are then included in the discussion in line 334:*

When looking at the night-time emissions, we find that per gram $CO_2$-C 0.06 g $CH_4$-C are emitted from ponds, and only 0.02 g $CH_4$-C from the semi-terrestrial tundra. This underlines again, that especially when considering thermokarst ponds, $CH_4$emissions are of high interest.

**Lines 271-272: "Our approach of combining a footprint model with a land cover classification to extract fluxes from different land cover classes allows us to determine the pond CO2flux." This sentence is in the core of all my difficulties in understanding what has**

**actually been done. Didn't you use the direct pond fluxes (from sector 60-120) to infer the pond CO2 flux?**
**At least this is what you mention on line 207, and in the table 1. And (in my understanding so far) you used the footprint model approach to estimate the flux from "tundra" (or semi-terrestrial tundra, vegetated tundra; are these same? If yes, please ease the reader's pain and use uniform expressions here. If not, please explain more clearly what's the difference.**

*Thank you for this helpful comment. We have included a new method section on CH4 flux in response to the other reviewer to clarify the difference between the analysis of $CO_2$ and $CH_4$ flux:*

**$CH_4$ flux partitioning**

Since we do not have a simple gap-filling model at hand for $CH_4$ emissions from the tundra, and since $CH_4$ emissions are much more variable than $CO_2$ emissions, we treat $CH_4$ differently. Instead of extracting the fluxes from the landcover types, we focus on wind sectors. We divide fluxes in wind sector:

- tundra: At least half of the footprint consists of dry tundra, and the wind direction is larger than 170°.
- $shore_{50°}$: Less than 40% of the footprint consists of dry tundra and water contributed to the footprint with at least 30%. The wind direction lies between 30° and 65°.
- pond: At least half of the footprint consists of open water and the wind direction lies between 65° and 110°.
- $shore_{120°}$: Less than 40% of the footprint consists of dry tundra and water contributed to the footprint with at least 30%. The wind direction lies between 110° and 130°.

**$CH_4$ permutation test**

To evaluate whether the differences in medians between the four wind sectors are significant, we apply a permutation test (Edgington and Onghena, 2007). In this test, we randomly assign each 30-min flux signal to one of two groups, calculate the median of both groups and their difference (with four groups it amounts to six tests in total). After repeating this step 10000 times, we plot the resulting differences in medians in a histogram and perform a one-sample t-test to evaluate whether the observed difference in medians differs significantly ($p < 0.01$) from the randomly generated differences.

*We also clarified in (the former) line 207 that the pond $CO_2$ flux was estimated using equation 4 from the method section. We hope that this makes our approach clearer. To improve readability, we added a sentence in the section of "Open-water $CO_2$ flux" that defines the semi-terrestrial tundra, and we now always only use this term, and not 'vegetated tundra'.*

Since we are interested in average tundra fluxes, we combine the landcover classes dry tundra, wet tundra, and overgrown water under the term *semi-terrestrial tundra*. In this way we can compare two landcover classes, semi-terrestrial tundra and the open water from thermokarst ponds.

**Chapter 4.2: The observation of the CH4 spike in the shore120 is interesting, and the fact that it remains unexplained, is pity but not unexceptional in flux studies! It is also**

**somewhat convincing how much effort you have had to explore the reasons for the higher emission**

*Thank you very much.*

**line 340: a somewhat similar approach has been used also earlier, see e.g.**
**https://bg.copernicus.org/articles/16/255/2019/**

*Thank you for this comment. In the method section we tried to embed our approach better into the existing literature and we now refer to this work and the work by Rößger et al. (2019a); Rößger et al. (2019b) by adding a new sentence:*

Similar approaches of analyzing heterogeneous eddy covariance fluxes in the arctic environment have been conducted for $CO_2$ and $CH_4$ (e.g. *Rößger et al. (2019a); Rößger et al. (2019b)*b; Tuovinen et al. (2019)). Rößger et al. 2019 a,b extracted $CO_2$ and $CH_4$ fluxes from two different land cover classes on a floodplain, and Tuovinen et al. (2019) separated $CH_4$ fluxes from nine different land cover classes, including water, and combined them into four source classes. All three studies have in common that they differentiate fluxes from different vegetation types, however our method is dedicated to differentiating between fluxes from tundra and water.

**Figures 2 and 5 (which are nice and indicative figures overall!): please indicate if the fluxes consist of purely measured values, purely gap-filled, or both. If just measured values are shown, how are the mean values (in red) used in the study? If there are missing values during the day, the mean does not represent the true daily NEE. Is the mean (red value) a mean of all fluxes from that direction during the 2-month period?**

*Thank you for this suggestion. We added the information about observed or modelled data in all plots and included the following words in Figure 2 and 5 to clarify that the 5° average was taken for the whole 2-months period:*

during the 2-months observation period

**Figure 3: is each dot a 30-min flux? Please explain it. "Flux intervals at night time"? Why interval, aren't these just fluxes?**

*Thank you for this comment. We think that confusion arises, because we used the term "flux interval" for 30-minute fluxes. In our view one 30-minute flux can be also defined as a flux interval. For clarification, we added the term "30-minute" in front of "flux" and removed the "interval". We also edited all other parts in the manuscript in this way.*

**Fig 4: please explain how the violin plot should be interpreted**

*Thank you for this suggestion. We added to following text at the end of Fig. 4 and 6:*

A violin plot shows the distribution of measurements along the y-axis - the width of the curves expresses the density of data points at each y-value.

**Fig A2: what is the red line?**

*We added to following in the figure caption:*

The observed differences in medians between the different wind direction classes are shown in red vertical bars in each plot.

**Figs A2 & A3 and line 310: there are no b's or c's in Fig. 6**

*Thank you for noticing. These letters were included in a previous version of the manuscript. We deleted the letters.*
* * *
Additionally, while working on the suggestions by the reviewers, we found a small error in the program code. We corrected for this error and updated the figures 6, A4, and A5 and updated the numbers on CH4 flux in table 1 and section 3.3. The error had only a small impact and had only a noticeable effect on the statistical test of figure A4, center graph. Now, the difference in medians between *pond* and *shore$_{50°}$* is significant and we described this in the discussion. We also updated the resulting numbers on *pond* CH4 flux (median *pond* CH$_4$-C flux before correction 13.38 mg m$^{-2}$d$^{-1}$, after correction 13.90 mg m$^{-2}$d$^{-1}$). We also had to update the standard deviation of CO2 flux at the beginning of the result section. However, all the conclusions we draw remain unchanged.

[Figure]

Updated Figure 6. Violin plots of observed CH4 emissions at the EC tower separated into four different wind direction classes. A violin plot shows the distribution of measurements along the y-axis - the width of the curves expresses the density of data points at each y-value. Medians of CH4 emission distributions are shown as red lines, and 75th & 25th percentile are shown as black lines. On the right, the wind sectors with the eddy covariance tower in the center (black cross) are shown.

[Figure]

Updated Figure A4. Histogram of permutation tests between the medians of CH4 emissions from different wind direction classes in figure 6. All medians from flux observations during moderate wind speed conditions. The observed differences in medians between the different wind direction classes are shown in red vertical bars in each plot.

[Figure]

Updated Figure A5. Histogram of permutation tests between the medians of $CH_4$ emissions from different wind direction classes in figure 6. All medians from flux observations during moderate air temperature conditions. The observed differences in medians between the different wind direction classes are shown in red vertical bars in each plot.

**Literature**

Edgington, E. and Onghena, P., 2007. Randomization tests. CRC press.

Rößger, N., Wille, C., Holl, D., Göckede, M. and Kutzbach, L., 2019a. Scaling and balancing carbon dioxide fluxes in a heterogeneous tundra ecosystem of the Lena River Delta. Biogeosciences, 16(13): 2591-2615.

Rößger, N., Wille, C., Veh, G., Boike, J. and Kutzbach, L., 2019b. Scaling and balancing methane fluxes in a heterogeneous tundra ecosystem of the Lena River Delta. Agricultural and Forest Meteorology, 266-267: 243-255.

Runkle, B.R.K., Sachs, T., Wille, C., Pfeiffer, E.M. and Kutzbach, L., 2013. Bulk partitioning the growing season net ecosystem exchange of $CO_2$ in Siberian tundra reveals the seasonality of its carbon sequestration strength. Biogeosciences, 10(3): 1337-1349.

Tuovinen, J.P., Aurela, M., Hatakka, J., Räsänen, A., Virtanen, T., Mikola, J., Ivakhov, V., Kondratyev, V. and Laurila, T., 2019. Interpreting eddy covariance data from heterogeneous Siberian tundra: land-cover-specific methane fluxes and spatial representativeness. Biogeosciences, 16(2): 255-274.

---

## Author Response (AR1)

Not accounting for thermokarst ponds leads to overestimation of tundra carbon uptake
Lutz Beckebanze, Zoé Rehder, David Holl, Charlotta Mirbach, Christian Wille, and Lars Kutzbach

Dear Andreas Ibrom,

thank you again for your very fast reply. We are happy to address your additional points.
Additionally, we also carefully screened the text and hope that language and style are improved.
Please see our point-by-point reply below.

Please clarify in the methods part, whether or not you filtered the data for wind direction before gap filling (I assume you did).

> We filtered for wind directions and added the following descriptions in the method section:

> *In the model input, we exclude $CO_2$ fluxes with an absolute value of more than 4 g $m^{-2}$ $d^{-1}$. We additionally exclude $CO_2$ fluxes from the wind direction (WD) of the merged polygonal pond (30°< WD <150°) from the training dataset to obtain a dataset consisting of as much semi-terrestrial tundra as possible.*

please comment on the bimodal nature of the probability distributions in Fig. 3. Is this due to two distinct seasonal behaviour?

> Unfortunately, we are not certain what you mean with bimodal behavior in Fig. 3. We spend some more time describing the distribution by adding the following sentence:

> *We also find that low air temperatures are mostly associated with low respiration rates.*

When you describe sectors, e.g. for the case ">30° & <150°" consider the notation 30°< WD <150°.

> This is a very good suggestion. We adopted the notation.

We hope that these changes are satisfactory.

Best regards,
Lutz Beckebanze and Zoé Rehder on behalf of the authors.

---

## Author Response (AR2)

Ignoring carbon emissions from thermokarst ponds results in overestimation of tundra carbon uptake

Lutz Beckebanze, Zoé Rehder, David Holl, Christian Wille, Charlotta Mirbach, and Lars Kutzbach

Dear Reviewer,

Thank you for your feedback. Please see our detailed responses below.

**Thank you for the revised manuscript. Nice job done. I think, however, that the manuscript still needs work to clarify the presentation. I found the text incoherent, in many cases. I suggest that the authors will put some effort to review the consistency of the used terminology (for instance, ponds and models) and the logic and order how the different modeling and upscaling steps appear in the text. I feel that the problem is the precision of description.**

**These include also not so precise use of "CO2 flux" (shouldn't it be net ecosystem exchange of CO2, or a component flux, with direction relative to the atmosphere).**

> Thank you very much. In terms of consistency, we streamlined the use of the following words: ponds (and their different categories), models, NEE, $CO_2$ flux. Regarding the $CO_2$ flux, we also added a general sentence in the beginning of section 2.3 to define the direction of the fluxes:

> *We performed the raw data processing and computation of half-hourly fluxes for open-path and enclosed-path fluxes $CO_2$, $CH_4$ and $H_2O$) using EddyPro 7.0.6 (Licor, 2019). **The convention of this software is that positive fluxes are fluxes from the surface to the atmosphere, while negative fluxes indicate a flux from the atmosphere downwards.***

> Regarding the modeling and upscaling steps, please see our comment below.

**And the handling of the CH4 data raises questions. One reason for my stumbling through the text may be in the end of the introduction: "Due to the tower's position, fluxes from the merged polygonal pond are the dominant source of the observed EC fluxes under easterly winds. The observed EC fluxes are dominated by semiterrestrial polygonal tundra with only a low influence from small thermokarst ponds from the other wind directions. We aim to deepen our understanding of carbon emissions from thermokarst ponds and constrain their impact on the landscape carbon balance. To this end, we (1) compare the water body and tundra fluxes focusing on temporal and spatial patterns, and we (2) investigate the influence of the merged polygonal pond on the landscape carbon balance."**
**That paragraph should precisely describe the objectives and questions. I may be wrong, but should you state here something like that your tower was positioned so that you can assess fluxes from the merged polygonal pond (large enough to get a pure 'thermokarst pond' signal) and polygon tundra composed of small ponds and terrestrial tundra (centers and rims?). Then, the objectives are, 1) use footprint model and 2) model net ecosystem CO2 exchange using the footprint weights of terrestrial tundra and thermokarst ponds (polygon ponds?), 3) examine temporal and spatial patterns of NEE & CH4 from terrestrial and aquatic tundra/land cover types, and 4) investigate the influence of the small thermokarst ponds (merged and polygon centers?) on the landscape NEE of CO2 (and CH4?) over the xxx period.**

Ignoring carbon emissions from thermokarst ponds results in overestimation of tundra carbon uptake

Lutz Beckebanze, Zoé Rehder, David Holl, Christian Wille, Charlotta Mirbach, and Lars Kutzbach

Thank you for this detailed suggestion. We agree with your suggestion and have adapted the introduction accordingly.

*This paper aims to deepen the understanding of carbon emissions from thermokarst ponds and constrain their impact on the landscape carbon balance.*

*We (1) examine the temporal and spatial patterns of NEE and the spatial pattern of $CH_4$ flux from semi-terrestrial tundra and thermokarst ponds, and (2) investigate the influence of the thermokarst ponds on the landscape NEE of $CO_2$ during the months June to September 2019. To this end, we use a footprint model and model net ecosystem exchange (NEE) of $CO_2$ using the footprint weights of semi-terrestrial tundra and thermokarst ponds.*

Additionally, we also changed the pond terminology in the introduction to clarify the differences between polygonal ponds, merged polygonal ponds and the more general term thermokarst ponds.

*These **thermokarst** ponds are often only as large as one polygon **(polygonal ponds)**. When several polygons are inundated, this can cause larger shallow **thermokarst** ponds to form, which we term merged polygonal ponds (Rehder et al. 2021).*

**Line numbering would help pointing the comments, but here just some examples from the text Usually the mat&met and results are written in past tense. Check some published papers for a reference. A language edit maybe.**

We apologize for forgetting to insert line numbers. We followed your suggestion and changed the mat&met section to the past tense.

**p. 4. Fig. 1 add somewhere that there are larger thermokarst lakes in the map (not included in the study)**

While some of the large water bodies visible on the orthophoto are indeed thermokarst lakes, it is not clear if all the lakes on Samoylov formed through thermokarst processes. However, we add a sentence pointing that we exclude the large lakes.

*The location of the study site in Russia is shown in (a) and the location of Samoylov Island within the Lena River Delta is shown in (b). Samoylov Island is shown in (c); the surrounding Lena River appears in light blue. The outline of the river-terrace land-cover classification (Sect. 2.4.1 is indicated by the blue line. **We focus on the polygonal tundra, however, large lakes are excluded (circled in yellow)**. In (d),...*

**p. 5. Add what's the area of upscaling somewhere in the mat&met**

We were unsure what's meant, we changed two things to be on the safe side:

1) We added the following part in the section of the land-cover classification:

*(3.0 km², area within the blue line in Fig. 1 c)*

and we also added a description in the section of the open water $CO_2$ flux:

*To evaluate the impact of thermokarst ponds on landscape $CO_2$ flux, we estimated a polygonal tundra landscape-$CO_2$ flux from the late-Holocene river terrace of Samoylov Island...*

2) We included the absolute area of the land-cover classification in the section of the land-cover classification (2.4.1) and added the relative contribution of each land-cover class in this section.

*The land-cover classification has a resolution of 0.17 m x 0.17 m. It is projected onto WGS 1984 UTM Zone 52N and the land-cover classes include open water **(15.7%)**, overgrown water **(7.0%)**, dry tundra **(65.1%)**, and wet tundra **(12.1%)**, as defined by Muster et al. (2012)*

**p. 6. There are some illogic in describing the models and missing equation references Explanation of the different grey shades is good in the figure legend, but no need in the main text.**

We agree and moved the explanation of the different gray shades to the figure caption.

**P. 6. Number the models according to their appearance in the text. i.e. NEE (1), ….**

Thank you for pointing this out. We re-structured the section and start with the description of the NEE followed by the description of the NEE components. Now the order of the equations fits the order in which we reference them. Please refer to section 2.4.3.

**p. 7. in addition to the R2, RMSE and mean absolute error would be great, because the R2 is not informative how close the absolute values are.**

Thank you for this suggestion. We now provide information about the overall model performance in the form of the RMSE for both applications of the bulk-NEE model:

*The final RMSE between the model input and the gap-filled NEE had a value of 0.29 g $m^{-2}d^{-1}$.*

And

*This gap-filling modeling of $CO_2$-C flux had an RSME of 0.31 g $m^{-2}d^{-1}$.*

**p. 7. Would it make any more sense to explain the extraction of pond values before the gap-filling?**

Thank you for this suggestion. While we would like to keep the general order, we moved the section about the detailed application to the first usage of the model to 2.4.4 to better separate the model description and the model application.

Now section 2.4.3. only explains the general model. In section 2.4.4. we apply this model twice. In this process, we also re-named the section 2.4.4 from "The open-water $CO_2$ flux" to "*Separating CO2 fluxes from tundra and open-water*".

Ignoring carbon emissions from thermokarst ponds results in overestimation of tundra carbon uptake

Lutz Beckebanze, Zoé Rehder, David Holl, Christian Wille, Charlotta Mirbach, and Lars Kutzbach

**And be precise if question is about the merged pond or polygon ponds or something else**
**Why it is worth mentioning that others compared between different vegetation types and here between tundra and water?**

> We clarify the aim of this step by adding the following sentence in section 2.4.4:

> *Since our aim was to assess the impact of thermokarst ponds (both polygonal ponds and merged polygonal ponds) on NEE, we needed to eliminate the influence of polygonal ponds from our NEE estimate.*

**p. 8. At hand? How the others gap-fil CH4 fluxes? Can't you use the data even without gap-filling if enough data? Give a rationale why these wind directions were chosen for the comparison, now pretty random in the text.**

> Thank you for pointing this out. We changed the beginning of the section to make clear why we can't use a gap-filling model and added information on why we chose the wind sectors:

> *The $CH_4$ emissions from the heterogeneous landscape around the tower are less spatially uniform than to the $CO_2$ emission. Therefore, we can-not use a gap-filling model for the $CH_4$ comparable to the bulk model we used for $CO_2$. Thus, we investigate $CH_4$ emissions differently. Based on preliminary results from our analysis and the aerial image of the study site, we focus on four wind sectors instead of extracting the fluxes from the landcover types.*

**p. 9. When comparing, variation is better to be expressed as the ratio of the standard deviation to the mean. It is coefficient of variation.**

> Since we think there is meaning in the absolute values, we use instead the median and the percentiles, which are also used several times in the manuscript. Our point is that the fluxes from the tundra reach higher positive and negative numbers, indicating a stronger diurnal cycle, which can also be seen in Fig. 2. However, we clarify that we are only comparing absolute values.

> *The $CO_2$-C fluxes from this pond sector show a smaller absolute variability ($0.09^{0.38}_{-0.33}$ g $m^{-2}d^{-1}$, $Median^{95\% Percentile}_{5\% Percentile}$) than the fluxes from all other wind directions ($-0.08^{0.87}_{-1.56}$ g $m^{-2}d^{-1}$, $Median^{95\% Percentile}_{5\% Percentile}$).*

**P. 9. In the permutation test section: Should there be something else than "one of the two groups" in the next sentence? pairs? "In this test, we randomly assign each 30-min flux to one of two groups and calculate both groups' median and their differences."**

> Thank you for this suggestion. We added "all possible combination of pairs" to the following sentence to make the pair-structure of the permutation test clearer. The new sentence reads:

> *We conducted six tests in total, using all possible combinations of pairs with the four wind sectors.*

Ignoring carbon emissions from thermokarst ponds results in overestimation of tundra carbon uptake

Lutz Beckebanze, Zoé Rehder, David Holl, Christian Wille, Charlotta Mirbach, and Lars Kutzbach

**p. 10. About the shore sections. "These peaks did not correlate with any of the four land-cover classes" is unclear**

> Thank you for this suggestion. We tried to clarify this sentence by changing it in the following way:

> *These peaks did not correlate with a specifically large contribution of one of the land-cover classes to the footprint.*

**p. 10. How about the high mean values in the sector 300-360? Don't neglect if 120 deg gets attention.**

> Thank you for this comment. We also had a look at the wind direction 290-330° and the peak at 180-190° in our analysis. However, the focus of this manuscript lies more on the water body fluxes than on spatial heterogeneity of tundra fluxes.  We will  focus on the tundra fluxes more in an upcoming manuscript which uses more data. By averaging over the tundra wind sectors, we can compare an 'average tundra' to the water body fluxes. Nevertheless, we are happy to include more information on the peak at 180°-190° instead, which we think is a more pronounced 'peak' and thus more comparable to 120° (see text below).

> *In summary, neither high wind speed nor high temperatures act as a driver for the high $CH_4$ emission from $shore_{120°}$. In contrast, the peak at 180°-190° can be explained reasonably well using air temperature and friction velocity in a multiple linear regression ($R^2 = 0.44$). Using the same predictors results in an $R^2$ of 0.20 for the peak at $shore_{120°}$.*

**Fig. 6. is really nice**

> Thank you!

**p.14. Comparison with the other pond studies in the area is a bit confusing. Methodological differences are possible but note also whether those were same or different ponds and how large the samples were.**

> Thank you for this comment. We added the number of samples (46) used by Abnizova et al. and also explained that the samples were taken on ponds not within the footprint of our study site (section 4.1).

**p. 14. Maybe specify that with emissions you mean NEE of CO2 (direction to the atmosphere)**

> Thank you for this comment. As mentioned in a previous response, we streamlined our use of the terms NEE, $CO_2$ flux and emission.

**Fig A3. Lake? Do you mean the merged pond?**

> Yes, we changed the word to "merged polygonal pond".